# pTINCR microprotein promotes epithelial differentiation and suppresses tumor growth through CDC42 SUMOylation and activation

Olga Boix[1], Marion Martinez[1], Santiago Vidal[2], Marta Giménez-Alejandre[1], Lluís Palenzuela[1], Laura Lorenzo-Sanz [3], Laura Quevedo [4], Olivier Moscoso[5], Jorge Ruiz-Orera [6], Pilar Ximénez-Embún [7], Nikaoly Ciriaco[5], Paolo Nuciforo [8], Camille Stephan-Otto Attolini [9], M. Mar Albà[10,11], Javier Muñoz [7], Tian V. Tian[12], Ignacio Varela [4], Ana Vivancos[13], Santiago Ramón y Cajal [5], Purificación Muñoz[3], Carmen Rivas [2,14] & María Abad [1] ✉

The human transcriptome contains thousands of small open reading frames (sORFs) that encode microproteins whose functions remain largely unexplored. Here, we show that *TINCR* lncRNA encodes pTINCR, an evolutionary conserved ubiquitin-like protein (UBL) expressed in many epithelia and upregulated upon differentiation and under cellular stress. By gain- and loss-of-function studies, we demonstrate that pTINCR is a key inducer of epithelial differentiation in vitro and in vivo. Interestingly, low expression of *TINCR* associates with worse prognosis in several epithelial cancers, and pTINCR overexpression reduces malignancy in patient-derived xenografts. At the molecular level, pTINCR binds to SUMO through its SUMO interacting motif (SIM) and to CDC42, a Rho-GTPase critical for actin cytoskeleton remodeling and epithelial differentiation. Moreover, pTINCR increases CDC42 SUMOylation and promotes its activation, triggering a pro-differentiation cascade. Our findings suggest that the micro-proteome is a source of new regulators of cell identity relevant for cancer.

In the last decade, improvements in genomic computational analyses, peptidomics and ribosome profiling have revealed that our genome contains thousands of small open reading frames (sORFs) located in previously assumed non-coding regions, whose translation produces a myriad of bioactive proteins that have been largely overlooked[1,2]. These small proteins, shorter than 100 amino acids, are called micro-proteins, micropeptides or SEPs (from sORF-encoded peptides). To date, only a subset of them have been functionally characterized, and they have been shown to play essential functions regulating a plethora of fundamental processes such as DNA repair, RNA splicing, calcium signaling, cell metabolism and tissue regeneration[3–11]. Importantly, mounting evidence suggests that microproteins have a role in stress response[12–15]. In fact, several studies have observed an increased translation of sORFs in response to different cellular stresses such as oxygen and glucose deprivation[16], viral and bacterial infection[17,18] and even tumor initiation[19].

In epithelial tissues, cell polarity plays a pivotal role in the acquisition and maintenance of the differentiated status and in the specialized functions of epithelial cells[20]. The establishment of epithelial cell polarity relies on several coordinated events, including the reinforcement of cell-to-cell adhesion[21,22] and the regulation of cytoskeleton organization and its associated downstream signaling pathways[23,24]. The small GTPase cell division control protein 42-homolog (CDC42) is considered an essential orchestrator of epithelium formation[25,26]. Given its role in the structure and remodeling of the actomyosin cytoskeleton, it has been widely connected with the establishment of epithelial cell polarity[27–29] and with the maturation of epithelial cell contacts[30,31], among other important processes of epithelial morphogenesis.

Conversely, loss of epithelial identity by dedifferentiation is considered a fundamental initial step in epithelial tumorigenesis[32,33], associated with tissue disorganization, uncontrolled growth and re-acquisition of mesenchymal and stem cell features[34,35]. Epithelial tissue homeostasis is constantly challenged by exposure to pro-mutational injuries and, thus, robust protective mechanisms against malignant transformation are crucial. In this context, compelling evidence have shown that after genotoxic or oncogenic stress, epithelial cells activate a terminal differentiation program that has been proposed as a safe-guard mechanism against oncogenic alterations[36–38]. Altogether, cell polarity and differentiation have been positioned as key tumor suppressive mechanisms and their disruption is considered a hallmark in epithelial cancers[33,39]. Identifying players that regulate and maintain epithelial cell identity is necessary to understand development, tissue homeostasis and cancer biology. Importantly, the relevance of microproteins in regulating cell identity has not been explored.

*TINCR* was described as a lncRNA upregulated during keratinocyte differentiation, and it has been shown to promote epithelial differentiation as an RNA molecule[40–42]. In this study, we report that *TINCR* encodes an evolutionary conserved ubiquitin-like protein (UBL) that we have named pTINCR, which regulates epithelial differentiation by enhancing the SUMOylation and activation of CDC42, and which acts as a tumor suppressor in epithelial cancers.

## Results

### pTINCR is a conserved microprotein encoded by TINCR lncRNA and expressed in epithelial tissues

To identify novel microproteins with a possible role in cell identity, we evaluated the coding potential of annotated lncRNAs known to be regulated during differentiation and in cancer. We used PhyloCSF, a comparative genomics algorithm that analyzes codon substitution frequencies across evolution to predict coding sequences[43]. In skin, our analyses pointed to *TINCR* as a good candidate, based on its enriched expression[44] and its high PhyloCSF score (Fig. 1a and Supplementary Table 1).

*TINCR* is a 3.7-kilobase lncRNA upregulated during the differentiation of keratinocytes and other epithelial cells[40–42]. We corroborated that *TINCR* is expressed in human and mouse skin, and also in other epithelia (Fig. 1b). Conservation analysis of the *TINCR* transcript showed a sORF of 264 bp (Fig. 1c). Ribosome profiling analysis in mouse skin (RibORF score ≥ 0.7) revealed that this sORF is translated into an 87 amino acid microprotein highly conserved across the tetrapoda taxa, which we named pTINCR (Fig. 1c, d). To confirm that pTINCR sORF is translated into a stable microprotein, we performed in vitro translation using the full-length *TINCR* lncRNA (ENSEMBL: ENST00000448587.5) in the presence of $^{35}$S-methionine (Fig. 1e, f), and obtained a peptide product of ~12 kDa. Of note, mutating the start codon of pTINCR impairs the translation of any detectable product (Fig. 1e, f). In addition, by re-analyzing mass spectrometry data from human organotypic skin cultures[45], we identified three different tryptic peptides corresponding to pTINCR, providing strong experimental evidence of pTINCR translation in skin (Fig. 1g and Supplementary Fig. 1A). Furthermore, we performed Western blot and immunohisto-chemical analyses using a custom polyclonal antibody against pTINCR and observed that pTINCR protein is expressed in skin and also in several other stratified (tongue, palate, esophagus, bladder, cervix and mammary gland) and simple (lung, stomach, uterus, sebaceous and sweat glands) epithelial tissues (Fig. 1h–j and Supplementary Fig. 1B). Of note, apart from the expected band (Fig. 1h, indicated by the arrow), pTINCR antibody detects by Western blot additional bands that may correspond to post-translational modifications of pTINCR, although we cannot discard other uncharacterized pTINCR isoforms. Finally, to determine pTINCR subcellular localization, we performed immunofluorescence experiments in HaCaT and MCF7 cells, two epithelial cell lines documented to express high levels of

*TINCR* lncRNA according to the Human Protein Atlas project. Endogenous pTINCR microprotein was detected in these cell lines localized mainly in the nucleus and at the cell-to-cell junctions (Fig. 1k). Altogether, we have demonstrated that *TINCR* lncRNA actually codes for pTINCR microprotein.

### pTINCR promotes epithelial differentiation in vitro and in vivo

To characterize the cellular and molecular functions of pTINCR, we cloned pTINCR sORF tagged with an HA epitope in a doxycycline-inducible lentiviral vector. We generated two different constructs placing the HA-tag either in the C-terminal (pTINCR-HA) or the N-terminal (HA-pTINCR) part of the microprotein. To minimize the possible effect of the tag on pTINCR, we introduced a flexible linker between the HA and the pTINCR sORF (Supplementary Fig. 2A and Supplementary Table 2). Moreover, to uncouple the function of *TINCR* lncRNA and pTINCR microprotein, we also developed a synthetic ORF (syORF) by mutating ~20% of pTINCR-HA nucleotide sequence, significantly changing the secondary structure of the RNA while producing the same protein (Supplementary Fig. 2A, B, and Supplementary Table 2). We have used the syORF throughout the manuscript to validate our main findings (see figure legends). Both C- and N- terminal constructs were detected upon transient expression in U2OS (Supplementary Fig. 2C, D) and displayed similar distributions, being detected in all cellular fractions but most prominently in the nucleus and at cell-to-cell junctions (Supplementary Fig. 2E, F), consistent with the endogenous localization of the protein (Fig. 1k).

The high expression of *TINCR* in skin prompted us to evaluate the role of pTINCR firstly in this context. We used the human keratinocyte HaCaT cell line, which allows to recapitulate keratinocytes differentiation by modulating the $Ca^{2+}$ concentration in the culture medium. Consistent with a previous report[40], we observed that *TINCR* mRNA expression was higher in differentiated (cultured under high $Ca^{2+}$ concentration) compared to non-differentiated HaCaT cells (cultured under basal/low $Ca^{2+}$ conditions) (Supplementary Fig. 2G). Our results show that pTINCR overexpression during HaCaT differentiation led to an increased upregulation of differentiation markers compared to the control cells (Fig. 2a, differentiation medium panels), indicating that pTINCR improves keratinocyte differentiation. Remarkably, pTINCR overexpression induced an increase in the expression of some differentiation markers even in low $Ca^{2+}$ conditions (Fig. 2a, basal medium panels), suggesting that pTINCR is sufficient to trigger differentiation events. In agreement, we observed that pTINCR overexpression per se (without changing the $Ca^{2+}$ concentration) switches actin cytoskeleton organization from a stress fiber disposition to a cortical pattern (Fig. 2b, second row), resembling the keratinocyte differentiation process (Fig. 2b, third row)[23,24,46,47]. Moreover, pTINCR overexpression induced the establishment of adherens and tight junctions similarly to high calcium conditions, as seen by the increase in E-cadherin and β-catenin (markers of adherens junctions) and ZO-1 (marker of tight junctions) at the cell membrane (Fig. 2c–e).

To further explore the role of pTINCR as a driver of epidermal differentiation, we performed an in vivo proof-of-concept experiment using a teratoma formation assay, which allows to study pluripotent stem cell differentiation in an unbiased manner. We inoculated mouse Embryonic Stem Cells (ESCs) modified to express pTINCR or not in the flanks of immunodeficient mice and observed that pTINCR expression led to reduced teratoma growth (Fig. 2f), consistent with more differentiated teratomas. Furthermore, histopathological examination of teratoma sections revealed a significant increase in skin differentiation within pTINCR-overexpressing teratomas, supported by the increase in keratin deposition (Fig. 2g and Supplementary Fig. 2H). Immunostaining experiments in control teratomas (with no pTINCR overexpression) showed endogenous pTINCR expression in skin. Some teratomas presented transitions from single to stratified epithelia and, remarkably, pTINCR expression notably increased in the stratified

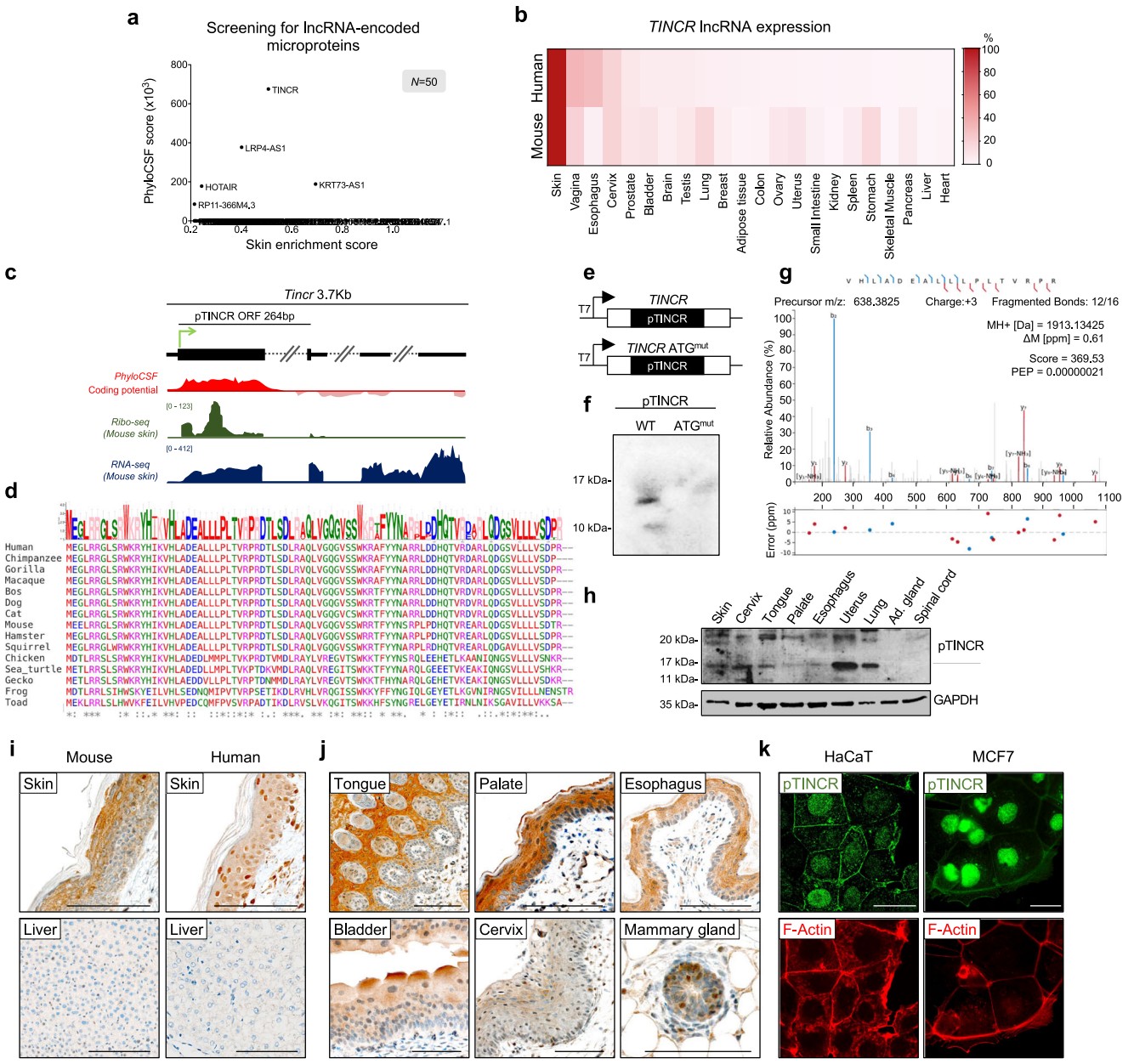

**Fig. 1 | pTINCR is an evolutionarily conserved microprotein highly expressed in epithelia. a** Screening of the coding potential of lncRNAs enriched in skin compared to other organs[44]. PhyloCSF score was obtained from LNCipedia v.5.2. **b** Heat map of *TINCR* lncRNA expression in human and mouse tissues. Expression in human was obtained from GTEX. Expression in mouse was assessed by RT-qPCR (at least 3 mice per organ). Values represent the percentage of *TINCR* expression in each organ normalized to the expression in the skin. **c** Diagram of *TINCR* locus. pTINCR sORF is indicated with a big black square. In red, PhyloCSF score across *TINCR*. In green, Ribo-seq analysis of *Tincr* transcript in mouse skin. In blue, expression of *Tincr* in mouse skin by RNAseq. **d** pTINCR amino acid conservation across the tetrapoda taxa depicted by multiple sequence alignment performed by Clustal Omega. Amino acid colors indicate their properties (pink, positive charge; blue, negative charge; red, hydrophobic; green, hydrophilic). The symbols below the alignment represent the biochemical similarity (asterisks indicate identical conservation, colons high similarity and periods similar conservation). **e** Diagram of the constructs used for in vitro translation of pTINCR microprotein. **f** In vitro translation of WT and ATG^mut full-length *TINCR* lncRNA visualized by SDS-PAGE. Two experiments were performed and obtained similar results. **g** MS/MS spectrum of a unique peptide derived from pTINCR microprotein in human skin organotypic cultures. **h** Western blot analysis of endogenous pTINCR in a panel of mouse epithelial tissues. Adrenal gland and spinal cord were added as negative controls. The experiment was repeated 4 times obtaining similar results. **i** Representative immunostainings of pTINCR in mouse and human skin and liver using a pTINCR antibody. Liver was added as a negative control. Scale bars correspond to 100 μm. **j** Representative immunostainings of a panel of mouse stratified epithelia using a pTINCR antibody. Scale bars correspond to 100 μm. **k** Representative immuno-fluorescence images of HaCaT and MCF7 cells using a pTINCR antibody (green) and Phalloidin-TRITC (red). Scale bars correspond to 50 μm. Immunostainings were performed at least twice obtaining the same results. Source data are provided as a Source Data file.

areas compared to the non-stratified ones (Fig. 2h and Supplementary Fig. 2l).

Next, we wanted to address the function of pTINCR in other epithelial contexts. We overexpressed pTINCR in a set of cancer cell lines of epithelial origin, including a patient-derived cutaneous squamous cell carcinoma (cSCC) cell line (hSCC10[48,49]), the A549 lung adenocarcinoma, and the MCF7 luminal breast cancer cell line. In hSCC10 and A549 cells, pTINCR overexpression was associated with an upregulation of several genes related to epithelial differentiation and a downregulation of basal epithelial markers

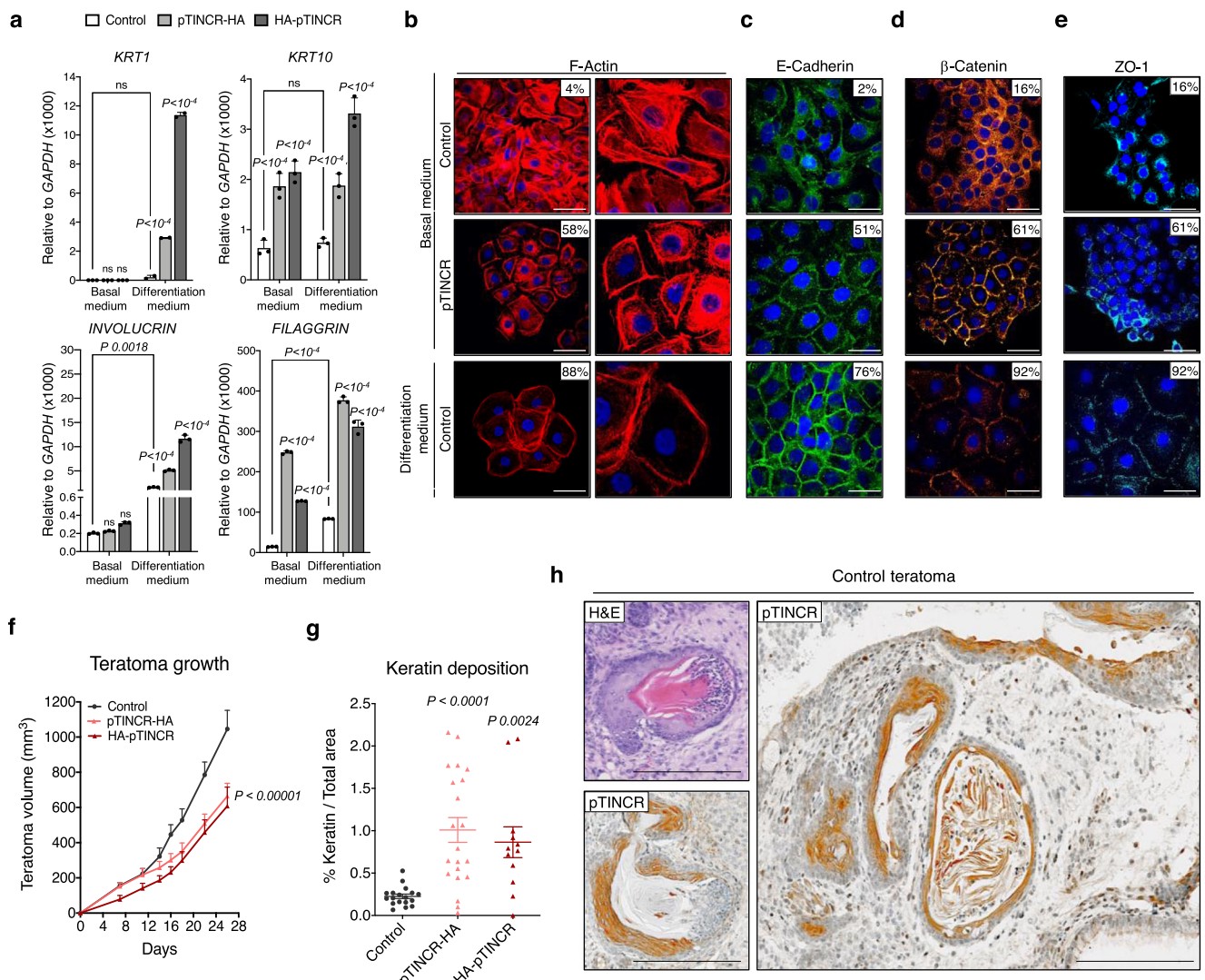

**Fig. 2 | pTINCR overexpression promotes epidermal differentiation.**
**a** Expression of the indicated differentiation markers in HaCaT cells upon pTINCR-HA (syORF) and HA-pTINCR overexpression. mRNA expression was measured 4 days after inducing pTINCR with doxycycline and changing to differentiation medium (or maintained in basal medium). Statistical differences between pTINCR-overexpressing cells and control cells (in basal or differentiation medium) and between basal and differentiated control cells (shown in brackets) are shown. Error bars represent the mean ± SD of $N = 3$ technical replicates from a representative experiment performed 3 times independently obtaining similar results. A 2-way ANOVA test corrected for multiple comparison was performed. **b** Immunostaining images of F-actin 24 h after doxycycline induction of pTINCR-HA (syORF) in HaCaT cells cultured in basal medium. Control cells cultured for 24 h in differentiation medium are shown as positive control. Percentage of cells showing cortical actin is indicated. The experiment was performed 3 times independently, representative images are shown. Scale bars correspond to 50 μm.
**c–e** Immunostainings showing E-cadherin (**c**), β-catenin (**d**) and ZO-1 (**e**) after 24 h (E-cadherin) or 4 days (β-catenin and ZO-1) of doxycycline induction of pTINCR-HA

(syORF) in HaCaT cells. Control cells cultured for 24 h in differentiation medium are shown as positive control. Percentage of cells showing membranous staining is indicated. The experiment was performed 3 times (c) or 2 times (d and e) independently, representative images are shown. Scale bars correspond to 50 μm.
**f** Effect of pTINCR overexpression on teratoma growth. Error bars represent the mean ± SEM. $N = 18$ teratomas in control group; $N = 20$ in pTINCR-HA group; $N = 11$ in HA-pTINCR group. One-way ANOVA test corrected for multiple comparison was performed. **g** Quantification of keratin deposition in differentiated teratomas using QuPath. Dots represent the percentage of area with keratin deposition relative to the total teratoma area. Error bars represent the mean ± SEM. $N = 18$ teratomas in control group; $N = 21$ teratomas in pTINCR-HA group; $N = 12$ teratomas in HA-pTINCR group. Non-parametric Kruskal–Wallis test corrected for multiple comparison was performed. **h** Representative H&E staining and IHC of endogenous pTINCR in a control teratoma. Pictures show single and stratified epithelia with keratin deposition. Scale bars correspond to 200 μm. Source data are provided as a Source Data file.

(Fig. 3a). In agreement, pTINCR-overexpressing MCF7 cells upregulated luminal markers and downregulated basal cell genes (Fig. 3a). As previously observed in HaCaT cells, pTINCR expression triggers a cortical actin disposition (Fig. 3b–d) and reinforces cell-to-cell contacts in these models, as seen by an increased accumulation of β-catenin, ZO-1 and E-cadherin at the cell membrane (Fig. 3b–d and Supplementary Fig. 2J, K).

Altogether, these results strongly support the role of pTINCR in promoting epithelial differentiation in vitro and in vivo.

## pTINCR is required for in vitro epithelial differentiation
To investigate the impact of pTINCR deficiency on cellular differentiation, we generated pTINCR-deficient cell lines by using

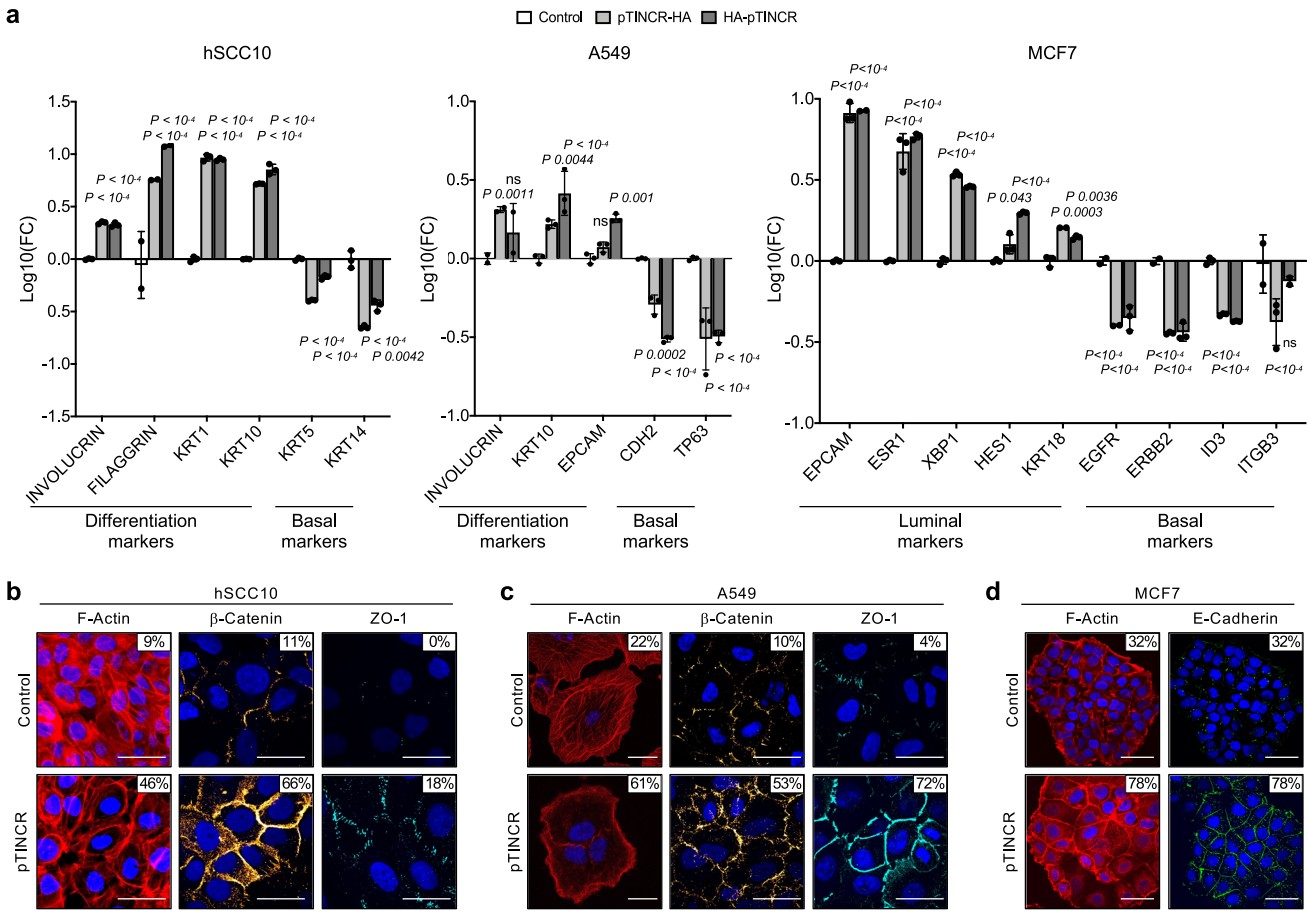

**Fig. 3 | pTINCR overexpression promotes differentiation in several epithelial cell types. a** Expression of the indicated differentiation and basal markers upon the overexpression of pTINCR-HA and HA-pTINCR in hSCC10, A549 and MCF7 cells. mRNA expression was measured 21 days (A549 and hSCC10) or 4 days (MCF7) after doxycycline induction. mRNA expression is normalized to GAPDH and relative to the control in each case. In MCF7, pTINCR-HA syORF was used. Error bars represent the mean ± SD of *N* = 3 technical replicates from a representative experiment performed at least 4 times independently obtaining similar results. A 2-way ANOVA test corrected for multiple comparison was performed.

**b–d** Immunostainings of F-actin, β-catenin, ZO-1 and E-cadherin in the indicated cell lines after 24 h (actin and E-cadherin) or 4 days (β-catenin and ZO-1) of doxycycline induction of pTINCR-HA (hSCC10 and A549) or pTINCR-HA syORF (MCF7). Nuclei are counterstained with DAPI. Percentage of cells showing membranous staining is indicated. Results observed upon the overexpression of HA-pTINCR are shown in Supplementary Fig. 2H, I. The experiment was performed twice independently, representative images are shown. Scale bars correspond to 50 μm. Source data are provided as a Source Data file.

CRISPR-Cas9 technology in two of our epithelial models, HaCaT and MCF7. pTINCR-KO cells contain a single-nucleotide insertion that disrupts the ATG start codon and abolishes the translation of pTINCR (Fig. 4a and Supplementary Fig. 3A–D), but is predicted to have no effect on the secondary structure of *TINCR* transcript (Supplementary Fig. 3E). We have validated pTINCR deficiency by Western blot (Fig. 4a) and immunostaining (Supplementary Fig. 3D).

We subjected both HaCaT and MCF7 cells to high calcium, a well-known inducer of differentiation in the epidermis[50] but also in other epithelial tissues such as esophagus[51] or mammary glands[52,53]. Of note, pTINCR protein expression increases upon differentiation in WT MCF7 cells (Fig. 4a). Importantly, we observed that *TINCR* lncRNA was upregulated upon calcium-induced differentiation to the same extent in WT and pTINCR-KO cells (Fig. 4b), indicating that the regulation of the *TINCR* lncRNA is not affected in engineered pTINCR-KO cells. However, pTINCR-deficient cells fail to upregulate differentiation markers to the same extent as WT cells (Fig. 4c, f). In addition, pTINCR-KO cells did not acquire an epithelial morphology (Fig. 4d, g) nor remodeled their actin cytoskeleton towards a cortical disposition under differentiation conditions (Fig. 4e, h). These results demonstrate

that pTINCR protein, independently from *TINCR* lncRNA, is required to achieve full epithelial differentiation in vitro.

**pTINCR triggers an epithelial differentiation transcriptional program**

To corroborate the pro-differentiation function of pTINCR, we analyzed the transcriptional profile induced by pTINCR overexpression. We performed an extensive RNA-seq analysis in hSCC10 cell line upon the overexpression of pTINCR to assess both early transcriptomic changes (6, 12, and 24 h) and long-term changes (4, 7, 14, and 21 days), the latter probably reflecting changes in cell identity. First, we studied transcriptional dynamics driven by pTINCR overexpression using impulseDE algorithm, a framework for longitudinal sequencing experiments that reveals differential gene expression associated with time[54]. We detected 6 different groups of genes that clustered together based on their dynamic expression (Fig. 5a, b and Supplementary Fig. 4A). We performed gene ontology enrichment analysis on each cluster (Fig. 5c, d and Supplementary Fig. 4B). Interestingly, two of them (cluster 1 and cluster 2) were dynamically opposite clusters, both enriched in gene ontology terms related to cytoskeleton. We used the

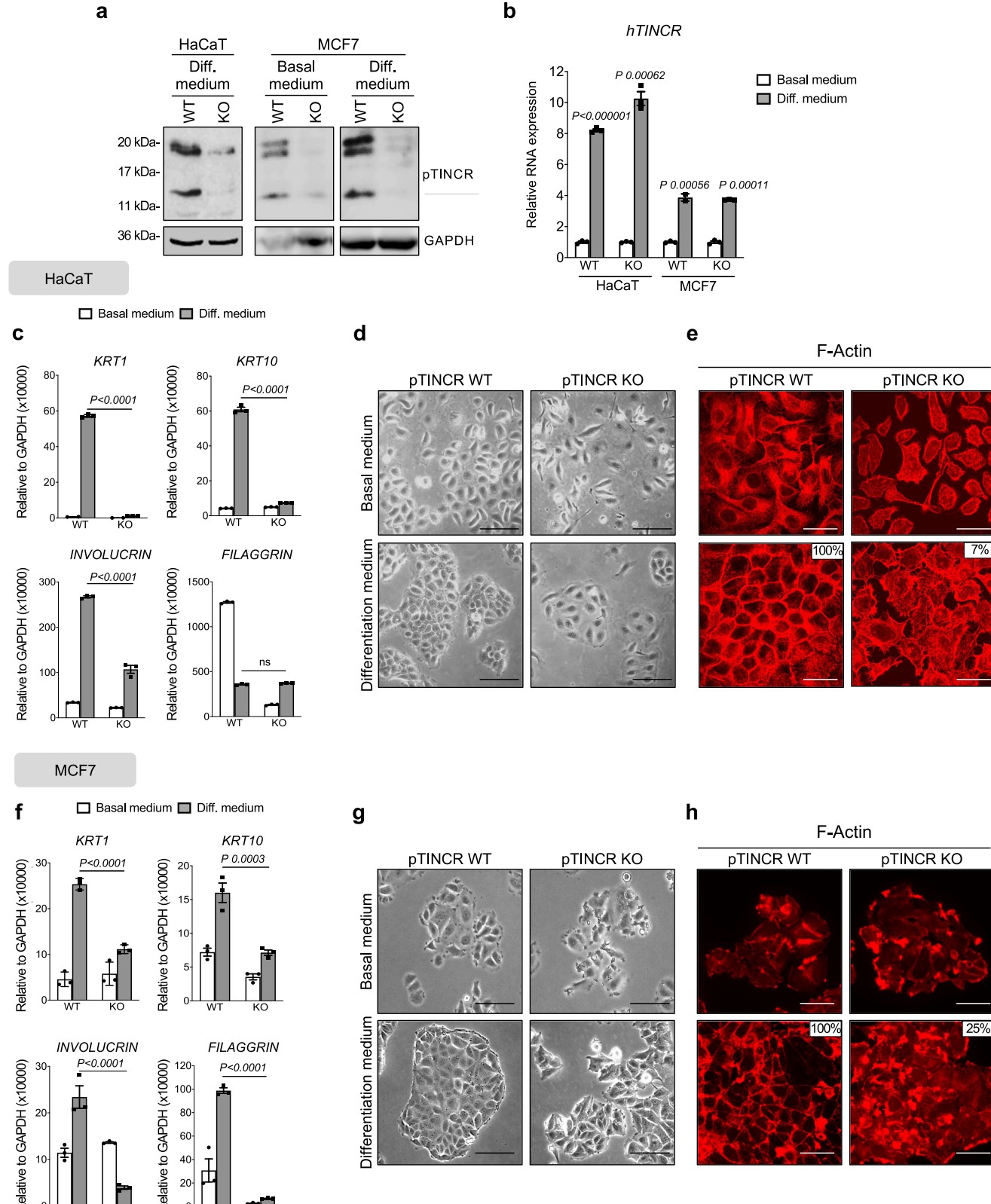

ClueGo tool to functionally organize gene ontology networks within each cluster (Fig. 5e, f). We observed that cluster 1 is enriched in genes connected with cell morphogenesis, cell-to-cell junctions and adhesion (Fig. 5e), and cluster 2 is formed by cytoskeleton organization-related genes associated with cell division and chromosome segregation (Fig. 5f).

Next, we ran DESeq2 analysis[55] in samples collected at 0 h and at 21 days and looked for long-term gene expression changes in pTINCR-overexpressing cells. A rotation-based functional enrichment analysis showed that gene-sets positively correlated with pTINCR expression were associated with actin organization and cell polarity, such as "actin filament organization", "establishment or maintenance of cell polarity"

**Fig. 4 | pTINCR is a critical regulator of epithelial differentiation in vitro.**
**a** Western blot of endogenous pTINCR using a polyclonal pTINCR antibody in WT and pTINCR-KO HaCaT and MCF7 cells, stably cultured in basal or high calcium differentiation medium, as indicated. **b** *TINCR* lncRNA expression measured by RT-qPCR in WT and pTINCR-KO HaCaT and MCF7 cells, under basal or differentiation conditions for 4 days. mRNA expression is normalized to GAPDH expression. Error bars represent the mean ± SD of *N* = 3 technical replicates from a representative experiment performed twice independently obtaining similar results. 2-way ANOVA test corrected for multiple comparison was performed. **c, f** Expression of the indicated differentiation markers measured by RT-qPCR in WT and pTINCR-KO HaCaT (**c**) and MCF7 (**f**) cells cultured in basal medium or high calcium differentiation medium for 4 days. mRNA expression is normalized to GAPDH expression. Error bars represent the mean ± SD of *N* = 3 technical replicates from a representative experiment performed twice (**c**) or 3 times (**f**) independently with similar results. n.s: not significant. 2-way ANOVA test corrected for multiple comparison was performed. **d, g** Representative phase contrast images of WT and pTINCR-KO HaCaT (**d**) and MCF7 (**g**) cells cultured in basal and differentiation conditions for 4 days. Scale bars correspond to 50 μm. **e, h** Representative immunofluorescence images of F-actin in WT and pTINCR-KO HaCaT (**e**) and MCF7 (**h**) cells cultured in basal medium or under differentiation conditions for 24 h. Total percentage of cells displaying cortical actin is indicated. Scale bars correspond to 50 μm. Source data are provided as a Source Data file.

or "cortical actin cytoskeleton" (Fig. 5g, h). In addition, other epithelia-specific gene sets ("Epithelial cell differentiation, "Cell–cell junction", "Epithelial cell-cell adhesion", "Apical surface", "Notch signaling") or epidermal-related ("Epidermal cell differentiation", "Skin development", "Keratinocyte differentiation", "Cornification") were also enriched after 21 days of pTINCR overexpression (Fig. 5g, h). On the other hand, the genes sets that negatively correlates with pTINCR expression were enriched in pathways associated with cell cycle, cell metabolism and protein processing, among others (Fig. 5g and Supplementary Fig. 4C). Interestingly, we found that pTINCR-downregulated genes were also enriched in the signature of "Myc targets" (Supplementary Fig. 4C), an oncogenic pathway in cSCC closely connected with the differentiation grade of tumors and clinical prognosis of this malignancy[44,56,57]. We validated these results by RT-qPCR analysis (Supplementary Fig. 4D, E). In conclusion, this transcriptomic study strongly supports the pro-differentiating function of pTINCR revealed by our functional studies.

**pTINCR is upregulated upon cellular stress in a p53 dependent manner and it is required for damage-induced differentiation**
p53 is widely known for its role as a tumor suppressor, being mutated in more than 50% of human cancers. In addition, mounting evidence has shown that p53 is also involved in embryonic development and cell differentiation[58,59]. Our previous results suggest that the role of pTINCR in calcium-induced differentiation does not require functional p53, given that HaCaT cells are p53 mutant. Given that in epithelial tissues cellular damage activates a terminal differentiation program[36,37], we wondered whether the expression of pTINCR was related to p53 activation upon damage. First, we treated different cancer cell lines with several p53 inducers or stabilizers (doxorubicin, actinomycin-D or nutlin-3a) and analyzed *TINCR* expression and pTINCR levels. We observed that only cells with functional p53 (A549 or HCT116) upregulated pTINCR at the mRNA and protein level, whereas p53-KO cells (HCT116 p53KO) or with mutant p53 (hSCC10) did not show the same response (Fig. 6a, b, Supplementary Fig. 5A–D). These results corroborated that pTINCR is upregulated upon stress in a p53 dependent manner. We analyzed published p53 ChIP-seq experiments[60,61] and we did not observe p53 binding to pTINCR locus, either in control cells or in cells treated with DNA-damaging agents, suggesting that the regulation of pTINCR by p53 is indirect (Supplementary Fig. 5E, F). Next, we studied the effect of pTINCR deficiency in p53-dependent DNA-damage-induced differentiation using the pTINCR-KO MCF7 cell line. MCF7 cells are p53 WT (Supplementary Fig. 5D) and, as expected, *TINCR* transcript was upregulated upon damage in both pTINCR-WT and pTINCR-KO cells (Fig. 6c). Consistent with previous results, this upregulation was abolished when MCF7 cells were transduced with the human papillomavirus (HPV) E6 oncoprotein, which induces p53 degradation (Fig. 6c and Supplementary Fig. 5D), and only WT MCF7 cells upregulated pTINCR at the protein level (Fig. 6d). Of note, we observed that pTINCR-KO MCF7 cells do not respond to damage-induced differentiation the same way as WT cells and fail to upregulate differentiation markers such as

FILAGGRIN and INVOLUCRIN, an effect that is partially restored when reexpressing exogenous pTINCR (Fig. 6e and Supplementary Fig. 5G). These results suggest that, upon cellular damage, pTINCR acts downstream the activation of p53 to induce cell differentiation.

**pTINCR has tumor suppressor activity in epithelial tumors**
*TINCR* lncRNA has been previously documented to be downregulated in human cSCC[40], where inactivating mutations in *TP53* are a well-known prevalent risk factor and an early event during the development of cSCC ([62], http://p53.free.fr). Our own analysis of transcriptomic data corroborated that *TINCR* expression is significantly decreased in cSCC compared to healthy skin (Fig. 7a). Moreover, we screened a panel of human patient-derived-cSCC cell lines (previously characterized by their epithelial or mesenchymal features[48,49]) for *TINCR* expression by RT-qPCR, and confirmed its downregulation when compared to the immortalized primary keratinocyte cell line HaCaT (Fig. 7b). Notably, epithelial-like cancer cell lines expressed higher levels of *TINCR* compared to mesenchymal-like cancer cell lines. Immunohistochemical analyses in a wide cohort of human cSCC tumors (*N* = 51) revealed that pTINCR expression is decreased in 50 out of 51 samples, where 40% of tumors have a > 50% reduction below normal levels (Fig. 7c, d, Supplementary Fig. 5H), supporting a tumor suppressive role of pTINCR in cSCC. The loss of pTINCR expression did not correlate with the mutational state of p53 in these tumors (Supplementary Fig. 5I, J), possibly meaning that tumors benefit from losing pTINCR expression independently of p53 mutation status. Consistent with these results, pTINCR overexpression significantly decreased tumor growth in PDXs generated with hSCC10 cells (Fig. 7e). Interestingly, pTINCR-overexpressing tumors presented a significantly higher deposition of hyaline matrix (Fig. 7f), a degenerative phenomenon associated with regression, DNA damage, autophagy, apoptosis and cell death[63]. The effect of pTINCR reducing tumor cell proliferation was also observed in hSCC10 in vitro and, importantly, in other epithelial cell types such as A549 and MCF7 (Supplementary Fig. 5K). Finally, we studied the potential tumor suppressor activity of pTINCR in other epithelial cancers and observed that high expression of *TINCR* significantly correlates with increased overall survival (OS) of patients with bladder carcinoma (Log rank test *p* = 0.00034, HR = 0.59), PDAC (Log rank test *p* = 0.0024, HR = 0.51), stomach adenocarcinoma (Log rank test *p* = 0.022, HR = 0.63), head and neck squamous cell carcinoma (Log rank test *p* = 0.0022, HR = 0.66) and lung adenocarcinoma (Log rank test *p* = 0.021, HR = 0.7) (Fig. 7g). Altogether, these results highly support the role of pTINCR as a tumor suppressor in cSCC and potentially in other epithelial tumors.

**pTINCR is an ubiquitin-like protein that interacts with SUMO and modulates SUMOylation**
To understand the molecular mechanisms behind the function of the microprotein, we did structural modeling using I-TASSER[64], which revealed that pTINCR is predicted to be a ubiquitin-like protein (Fig. 8a). Ubiquitin-like proteins (UBLs) are a family of proteins with structural similarity to ubiquitin, despite presenting low similarity in their amino

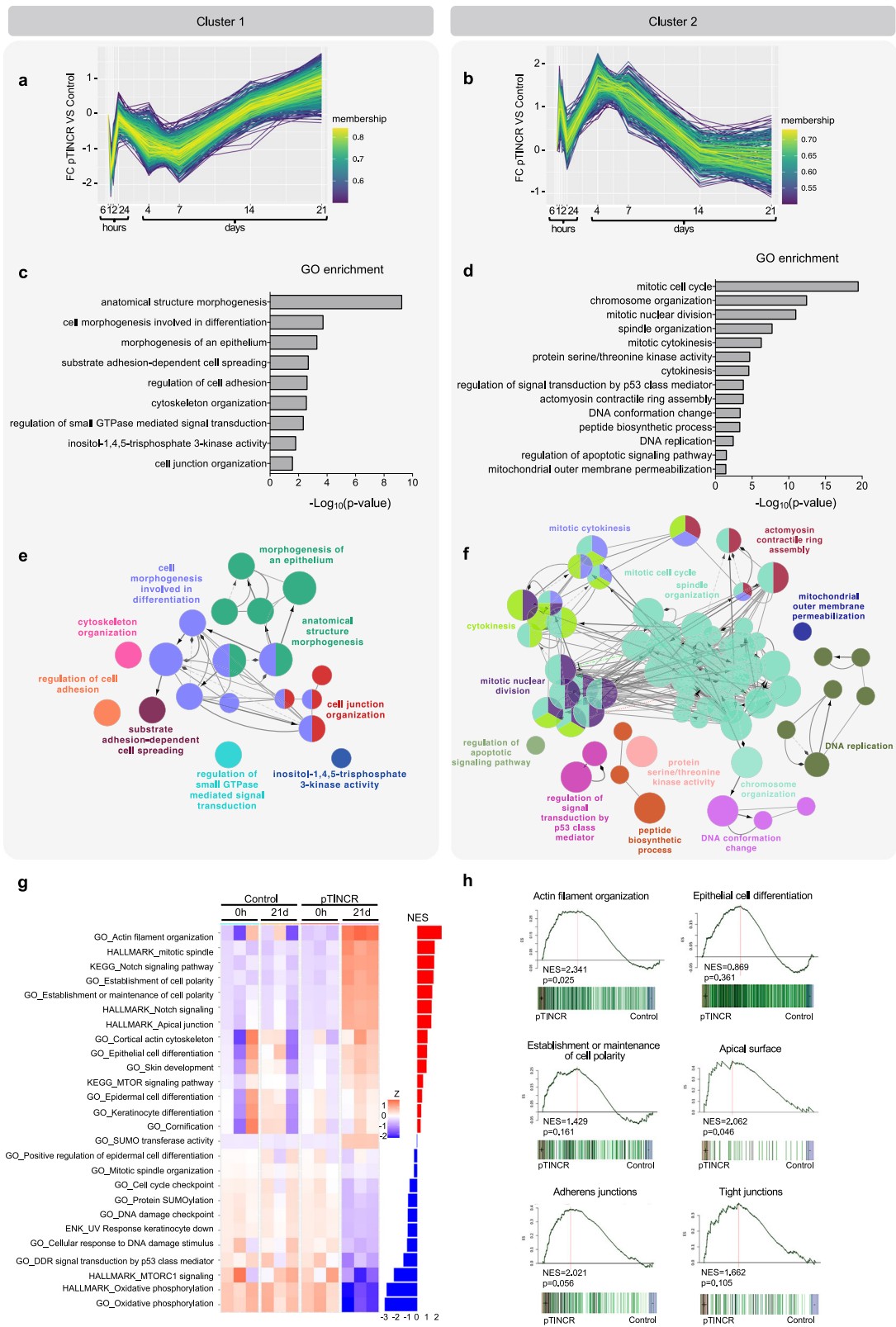

acid sequences[65]. Type I members of the UBL family contain a di-glycine (di-Gly) motif at the C-terminus for targeted conjugation to other proteins, functioning as covalent modifiers in a similar manner as ubiquitin[65]. On the other hand, Type II members of this family lack the di-Gly motif but bear one or more domains resembling ubiquitin structure[66,67]. Type II UBLs are not conjugated covalently to other molecules but can bind to their interactors and form protein complexes[67]. Therefore, given the absence of di-Gly motif, pTINCR could arise as a new Type II UBL (Supplementary Fig. 6A). Subsequent analysis of the pTINCR amino acid sequence using GPS-SUMO software[68,69] highlighted two well-conserved and overlapping SUMO-interacting Motifs (SIM) in the C-terminal part of the microprotein (Fig. 8a). To test the possible interaction of pTINCR with SUMO, we generated a SIM mutant version of pTINCR-HA (pTINCR-SIMmut) in

**Fig. 5 | pTINCR triggers transcriptional changes associated with actin cytoskeleton, epithelial morphogenesis and cell cycle.** RNA-seq analysis was performed in hSCC10 cells upon the overexpression of pTINCR-HA syORF to assess both early transcriptomic changes (6, 12, and 24 h) and long-term changes (4, 7, 14, and 21 days). **a, b** Transcriptional dynamics were analyzed using impulseDE, and genes were classified in clusters according to their similar expression dynamics. Graphs represent cluster 1 (**a**) and 2 (**b**). **c, d** GO terms enrichment analysis using ClueGO software indicating the functional term enrichments of the genes in clusters 1 (**c**) and 2 (**d**). Plots show the most significantly enriched GO terms. $p \leq 0.05$, using two-sided hypergeometric test with Bonferroni correction. **e, f**. GO enrichment analysis of genes from cluster 1 (**c**) and 2 (**d**) using ClueGO 2.5.1. Each node

represents a different GO term and its size is proportional to the Bonferroni-corrected $p$-value ($p \leq 0.05$ were considered significant). Significant GO terms are grouped based on their similarity (κ value) and labeled in the same color. Connected nodes shared common genes. Most significant GO terms in each group are labeled. **g** Heatmap of selected biological pathways affected by pTINCR overexpression, showing gene enrichment values of each experimental group at 0 h and 21 days of doxycycline induction. BarPlot shows normalized enrichment scores (NES) for the corresponding categories using GSEA. **h** Gene Set Enrichment Analysis of pTINCR-induced genes versus the indicated gene signatures. Source data are provided as a Source Data file.

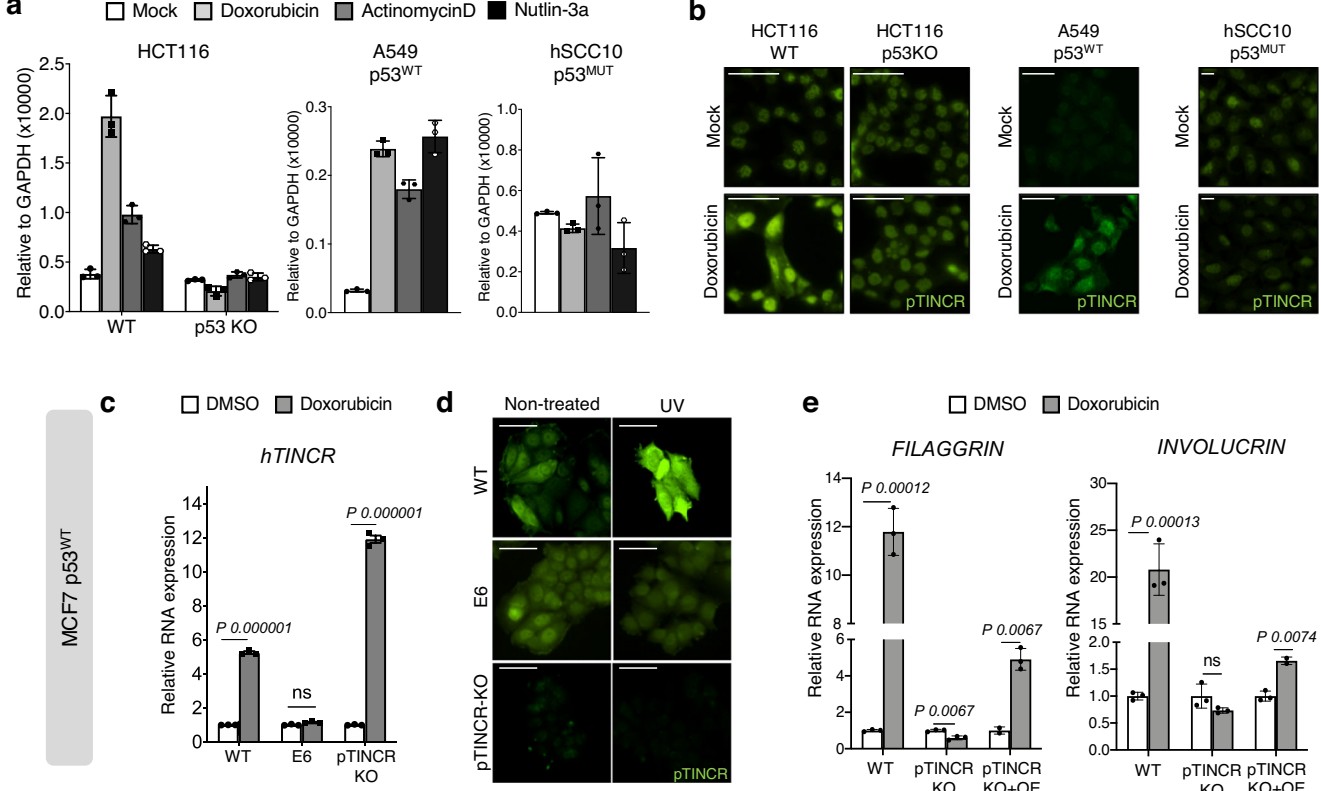

**Fig. 6 | pTINCR is induced by damage in a p53 dependent manner and it is required for damage-induced differentiation. a** *TINCR* lncRNA expression analyzed by RT-qPCR after 24 h of treatment with doxorubicin, actinomycin D or nutlin-3a in the indicated cell lines. mRNA expression is normalized to GAPDH in each case and relative to the control. Error bars represent the mean ± SD of N = 3 technical replicates from a representative experiment performed twice (hSCC10 and HCT116) or 3 times (A549) independently with similar results. **b** Endogenous pTINCR expression analyzed by immunofluorescence in the indicated cell lines after 24 h of treatment with doxorubicin. Representative images are shown, the experiment was performed twice independently. Scale bars correspond to 50 μm. **c** *TINCR* lncRNA expression levels analyzed by RT-qPCR after 24 h of doxorubicin in WT, E6-transduced or pTINCR-KO MCF7 cells. mRNA expression is normalized to GAPDH in each case and relative to the control. Error bars represent the mean ± SD of N = 3 technical replicates from a representative experiment performed 3 times

independently with similar results. n.s: not significant. Multiple T-TEST corrected for multiple comparison was performed. **d** Endogenous pTINCR expression analyzed by immunofluorescence after 24 h of exposure to UV light in WT, E6-transduced or pTINCR-KO MCF7 cells. Representative images are shown, the experiment was performed twice independently. Scale bars correspond to 50 μm. **e** Expression of the indicated differentiation markers after 24 h of treatment with doxorubicin in WT, E6-transduced, pTINCR-KO or pTINCR-KO re-expressing pTINCR-HA (pTINCR KO + OE) MCF7 cells. mRNA expression is normalized to GAPDH and relative to the control in each case. Error bars represent the mean ± SD of N = 3 technical replicates from a representative experiment performed 3 times independently with similar results. n.s: not significant. Multiple T-TEST corrected for multiple comparison was performed. Source data are provided as a Source Data file.

which most of the amino acids constituting the SIM domain were replaced by alanines (VLLLV > AAAAV, Supplementary Table 2). GST-pull down assays confirmed that pTINCR binds to SUMO1 and SUMO2/3 in a non-covalent manner and that the interaction is lost in the pTINCR-SIMmut (Fig. 8b). Interestingly, SIM-mutant pTINCR resulted in a less stable protein, as shown by cycloheximide treatment experiments, pointing to an important role of SUMO in stabilizing pTINCR (Fig. 8c).

Accordingly, pTINCR-SIMmut does not reduce cell proliferation and does not induce the remodeling of the actin cytoskeleton (Fig. 8d, e). To understand the meaning of pTINCR-SUMO interaction we analyzed global protein SUMOylation in different cell lines by Western blot. We observed that pTINCR overexpression modifies SUMO1 and SUMO2/3 conjugation patterns in a cell-dependent manner, increasing the intensity of some SUMOylated bands and decreasing the intensity of

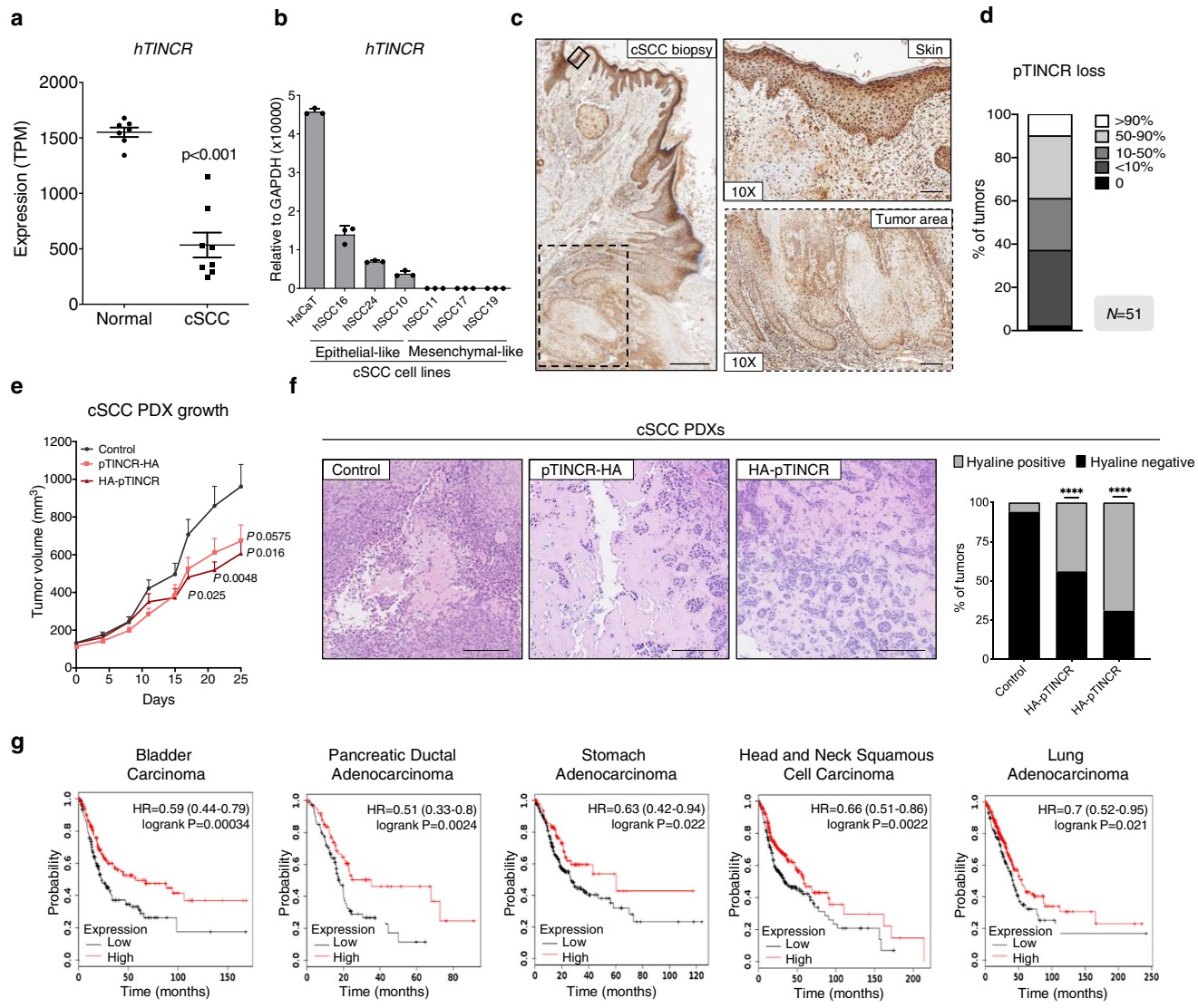

**Fig. 7 | pTINCR acts as a tumor suppressor in several epithelial tumors.**
**a** Expression of *TINCR* lncRNA in cSCC tumors compared to healthy skin (data subtracted from TCGA data). Statistical significance was calculated using two-sided T-TEST. **b** *TINCR* lncRNA expression in a panel of patient-derived cSCC cell lines and in HaCaT keratinocytes. Cancer cell lines are grouped according to their epithelial/mesenchymal traits. mRNA expression is normalized to GAPDH in each case. Error bars represent the mean ± SD of $N = 3$ technical replicates from a representative experiment performed twice independently with similar results. **c** pTINCR expression was analyzed by IHC in a cSCC patient cohort ($N = 51$). Representative images show the loss of pTINCR expression from healthy epidermis to cSCC (left) and a magnification of healthy skin and tumor area is shown (right). Scale bars correspond to 500 μm (left) or 200 μm (right). **d** Quantification of the

percentage of pTINCR loss in the cSCC patient cohort ($N = 51$). **e** Effect of pTINCR expression on cSCC in vivo tumor growth. Tumor growth was monitored by measuring the volume at the indicated time points. Error bars represent the mean ± SEM. $N = 16$ tumors in control group; $N = 15$ tumors in pTINCR-HA group; $N = 16$ tumors in HA-pTINCR group. Two-sided T-TEST was performed. **f** Representative images of H&E stainings showing areas with hyaline deposition in control and pTINCR-overexpressing cSCC PDXs. Graph shows the quantification of the number of tumors with hyaline deposits. Two-sided Fisher exact test was performed. Scale bars correspond to 200 μm. **g** Kaplan–Meier curves showing the correlation between *TINCR* expression and patient survival in the indicated tumor types. Source data are provided as a Source Data file.

others, an effect that seemed to be partially reverted in pTINCR-SIMmut cells (Fig. 8f, g and Supplementary Fig. 6B and C). Altogether, our data indicate that pTINCR is a UBL whose interaction with SUMO is essential for its stability and its function.

### pTINCR binds CDC42, enhances its SUMOylation and promotes its activation

Given that pTINCR binds to SUMO and, therefore, potentially interacts with many other SUMOylated proteins, we wanted to uncover its interactome. We performed immunoprecipitation of pTINCR-HA followed by mass-spectrometry analysis and detected 35 potential interactors with a Saint probability >0.7 and a Confidence score >25 (Supplementary Table 3). Among them, we were particularly interested

in CDC42 (Saint probability=1; FC-B > 6, Confidence Score=35) (Fig. 9a). CDC42 is a small GTPase of the Rho family, whose main function is to regulate the organization of the actin cytoskeleton[27]. Rho GTPases reversibly transit between an active (GTP-bound) and an inactive (GDP-bound) state, acting as molecular switches to regulate many essential cellular processes such as cell morphology, migration, invasion, cell cycle progression, endocytosis and gene transcription[70]. Moreover, CDC42 has been shown to be essential for epithelial morphogenesis by regulating cell-to-cell contacts formation and the establishment of apico-basal structures[28,71]. Interestingly, our RNAseq analysis pointed to a statistically significant positive correlation between pTINCR expression and the gene sets "GTPase binding" and "GTPase activator activity" (Fig. 9b). Thus, we wanted to explore

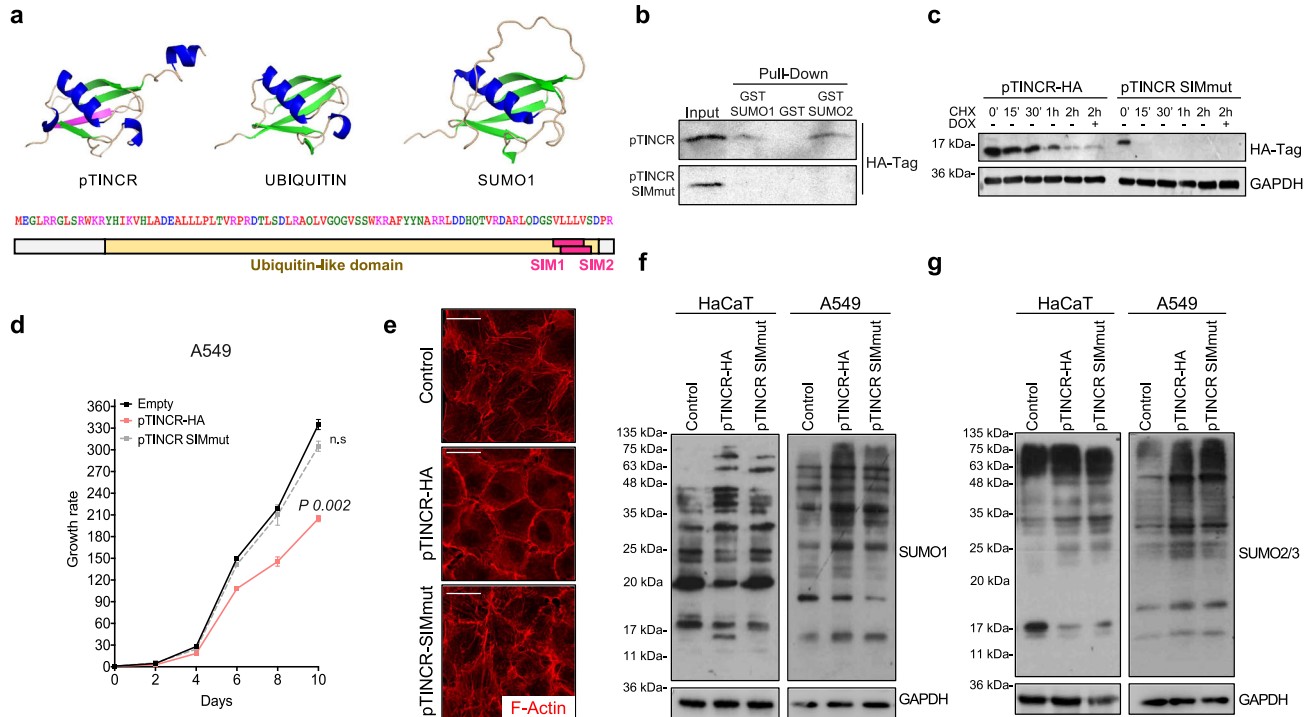

**Fig. 8 | pTINCR is a UBL that interacts with SUMO and modulates SUMOylation.**
**a** Upper panel, image depicting the predicted structure of pTINCR, UBIQUITIN and SUMO using PyMOL. Color code indicates: α-helix (dark blue); β-sheet (green); SIMs (purple). Lower panel, scheme of the predicted pTINCR domains. **b** Analysis of pTINCR and pTINCR-SIMmut non-covalent interaction with SUMO1 and SUMO2 by GST pull down assay. The experiment was performed 3 times independently obtaining the same result, a representative experiment is shown. **c** Analysis of pTINCR-HA and pTINCR-SIMmut stability in U2OS. Cells were treated with CHX for the indicated time points and analyzed by Western blot. The experiment was performed 3 times independently obtaining the same result. A representative experiment is shown. **d** Growth curve of A549 overexpressing pTINCR-HA, pTINCR-SIMmut and control vector. Growth rate represents cell number at each

time point relative to the starting number of cells (day 0). Error bars represent the mean ± SD of $N = 3$ technical replicates from a representative experiment performed 3 times independently obtaining similar results. n.s: not significant. Two-sided T-TEST was performed. **e** Effect of pTINCR-HA and pTINCR-SIMmut overexpression on actin cytoeskeleton in A549 cells. Panels show representative immunofluorescence images of F-actin in control, pTINCR and pTINCR-SIMmut overexpressing cells induced with doxycycline for 4 days. Scale bars correspond to 50 μm. **f, g.** Effect of pTINCR overexpression on SUMO1 (**f**) and SUMO2/3 (**g**) levels and conjugation analyzed by Western blot. In HaCaT cells, pTINCR-HA syORF was used. The experiment was performed 8 times in (**f**) and 3 times in (**g**) independently. A representative experiment is shown. Source data are provided as a Source Data file.

whether pTINCR pro-differentiation phenotype was related to CDC42. First, we validated the interaction between pTINCR and CDC42 by immunoprecipitation (Fig. 9c). Of note, we were not able to co-precipitate CDC42 with pTINCR-SIMmut. This could suggest that the binding of pTINCR to SUMO facilitates its interaction with CDC42, although we cannot discard the instability of pTINCR-SIMmut as an alternative explanation. Next, we assessed the effect of pTINCR on the activation status of CDC42 by pull-down experiments and observed that pTINCR, but not pTINCR-SIMmut, increases GTP-CDC42 levels (Fig. 9d). UBL proteins are characterized by their common, evolutionarily conserved tertiary structure, generally referred to as the β-grasp fold. This allows them to behave as adaptors for protein-protein interaction to ensure enzymatic activity[72]. Thereby, we hypothesized that the interaction of pTINCR with SUMO and its ability to activate CDC42 could be connected. First, we observed that CDC42 is SUMOylated and that this modification is enhanced when we overexpressed pTINCR (Fig. 9e, f). Worthy to mention, overexpressing pTINCR did not alter the SUMOylation status of other SUMO targets such as p53 and B23 (Fig. 9g), suggesting again that pTINCR effect on SUMOylation is not a global event but rather protein-specific. To our knowledge, the SUMOylation of CDC42 has not been described before. As seen by experiments using the SUMOylation inhibitor ML792 (Supplementary Fig. 6D, E), CDC42 SUMOylation did not significantly modify CDC42 localization (Supplementary Fig. 6F) or protein stability, given that we only see effects beyond 24 h of cicloheximide treatment (Supplementary Fig. 6G). However, the inhibition of

SUMOylation did decrease the levels of GTP-CDC42 in control cells and partially reverted pTINCR-induced CDC42 activation (Fig. 9h). These results are consistent with a previous report showing that the SUMOylation of Rac1, another Rho GTPase, does not modify its localization but is required for optimal GTP loading[73]. Altogether, these evidences suggest that pTINCR promotes CDC42 activation by enhancing its SUMOylation.

## pTINCR-induced CDC42 activation does not promote pro-oncogenic events
The epidermal growth factor (EGF) is a well-known CDC42 activator that is associated with the acquisition of migration and invasion capacities[74]. Importantly, although pTINCR induced CDC42 activation similarly to EGF cell treatment (Supplementary Fig. 7H), the outcome of the activation was different. While pTINCR triggered an actin cytoskeleton remodeling towards a cortical disposition (Figs. 2b, 3b–d), EGF led to the formation of lamellipodia and cell protrusions (Supplementary Fig. 7A). In agreement, pTINCR overexpression had no relevant effect on the transcriptional regulation of common epithelial-to-mesenchymal genes (Supplementary Fig. 7B, C), nor on cell migration or invasion (Supplementary Fig. 7D–G).

## pTINCR promotes epithelial differentiation through the activation of CDC42
Next, we wanted to assess whether pTINCR pro-differentiation phenotype was mediated by its ability to activate CDC42. We analyzed the

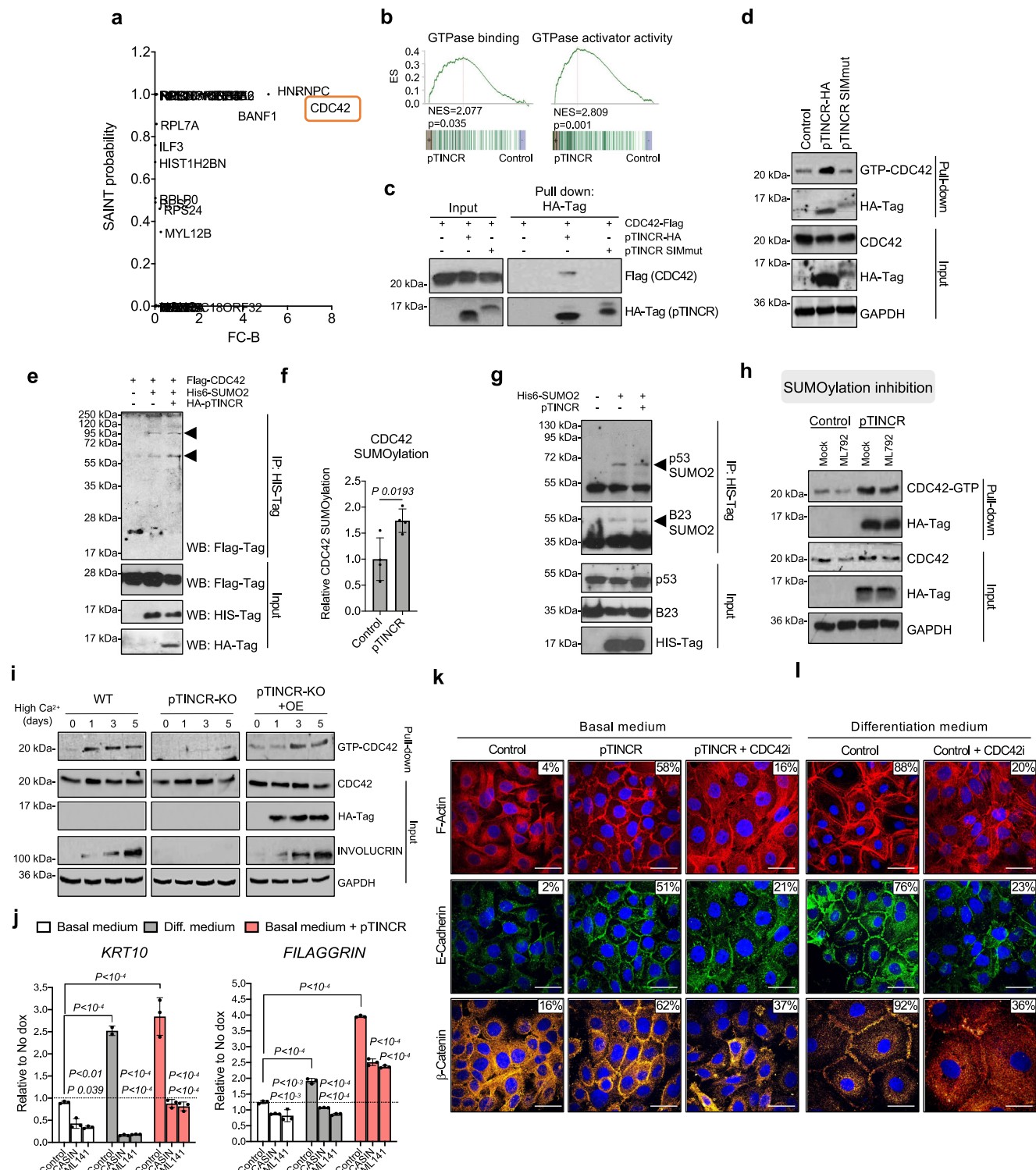

activation of CDC42 during calcium-induced differentiation in WT, pTINCR-KO and pTINCR-KO HaCaT cells transduced with pTINCR syORF (pTINCR-KO + OE). As observed in Fig. 9i, CDC42 is activated in WT cells during calcium-induced differentiation. This result agrees with previous reports showing that CDC42 activity is critical for processes associated with epithelial morphogenesis, such as cell junction maturation[25,30]. By contrast, pTINCR-KO cells failed to activate CDC42 and to differentiate (measured by the expression of INVOLUCRIN) upon exposure to differentiation conditions, despite having a functional *TINCR* lncRNA (Fig. 4b). Remarkably, this effect was partially reverted upon pTINCR re-expression in pTINCR-KO cells (pTINCR-

KO + OE), demonstrating that it is pTINCR microprotein and not the lncRNA that is responsible for CDC42 activation during differentiation. To further confirm that pTINCR promotes differentiation by inducing the activation of CDC42, we used two chemical inhibitors of CDC42 activation, CASIN and ML141, which block GTP loading onto CDC42 (Supplementary Fig. 7H). Of relevance, treatment with CASIN or ML141 inhibits both calcium- and pTINCR-induced differentiation, as seen by an impaired upregulation of differentiation markers (Fig. 9j and Supplementary Fig. 7I). Moreover, CDC42 inhibition also impaired the remodeling of actin cytoskeleton and the increase in cell-to-cell junctions formation induced by pTINCR (Fig. 9k) as well as by calcium-

**Fig. 9 | pTINCR promotes differentiation through the SUMOylation and activation of CDC42. a** Analysis of the pTINCR interactome using CRAPome algorithm. **b** GSEA analysis of pTINCR-induced genes in hSCC10 cells versus the indicated gene signatures. **c.** Interaction of pTINCR with CDC42 analyzed by co-immunoprecipitation assay. **d** CDC42 activation assay in hSCC10 cells after 4 days of pTINCR expression. pTINCR-HA syORF was used. **e** Representative CDC42 SUMOylation assay in U2OS cells transduced with pTINCR-HA, Flag-CDC42 and His6-SUMO2. SUMOylated CDC42 is indicated with arrowheads. **f** Quantification of 4 independent CDC42 SUMOylation assays performed in U2OS cells. Values of CDC42 SUMOylation are normalized to total CDC42 and relative to the control. Multiple T-TEST corrected for multiple comparison was performed. **g** Representative SUMOylation assay of endogenous p53 and B23 proteins in U2OS cells transfected with His6-SUMO2 and pTINCR-HA. SUMOylation of p53 and B23 is indicated with arrowheads. **h** CDC42 activation assay in HaCaT cells after 16 h of ML-792 treatment and/or 24 h of pTINCR syORF overexpression. **i** CDC42 activation assay in WT, pTINCR-KO or pTINCR-KO overexpressing pTINCR-HA syORF (pTINCR

KO + OE) HaCaT cells during calcium-induced differentiation. INVOLUCRIN is shown as a marker of cell differentiation. **j** Expression of the indicated differentiation markers after 24 h of calcium-induced-differentiation or pTINCR-HA syORF overexpression in HaCaT cells, treated or not with the CDC42 inhibitors CASIN or ML-141. Error bars represent the mean ± SD of $N = 3$ technical replicates from a representative experiment performed 3 times independently with similar results. We show statistical differences within each experimental condition and between basal control cells and differentiated control cells or pTINCR-overexpressing cells (shown in brackets). n.s: not significant. 2-way ANOVA test with multiple comparison was performed. **k–l** Immunostainings of F-actin, E-cadherin and β-catenin upon pTINCR- (**k**) or calcium-(**l**) induced differentiation for 24 h (actin and E-cadherin) or 4 days (β-catenin), in basal HaCaT cells treated or not with CASIN (actin and E-cadherin) or ML-141 (β-catenin). pTINCR-HA syORF was used. Nuclei are counterstained with DAPI. Total percentage of cells showing membranous staining is indicated. Scale bars correspond to 50 μm. Source data are provided as a Source Data file.

induced differentiation (Fig. 9l). These results demonstrate that pTINCR microprotein promotes epithelial differentiation, at least in part, by triggering CDC42 activation (Fig. 10).

To better define the molecular mechanism behind the pro-differentiation function of pTINCR, we performed an antibody-based array to analyze the phosphorylation status of 141 proteins associated with actin dynamics upon pTINCR overexpression (Supplementary Fig. 7J and Supplementary Table 4). Among the proteins with increased phosphorylation levels, we found an enrichment in members of the PKC, PLC and the phosphatidylinositol lipid (PtdIns) families, such as phospho-Ser307 PIP5K or phospho-Thr638 PKC α/β II (Supplementary Fig. 7K). This group of proteins and their effectors have been widely related to epithelial morphogenesis and CDC42 protein activation[25,75,76]. In contrast, several members of the MAPK family, whose activation is usually connected with pro-oncogenic and pro-proliferative outcomes, displayed decreased phosphorylation levels (Supplementary Fig. 7L). It is also worthy to mention that with this approach we detected a striking increase in F-ACTIN levels in pTINCR-overexpressing cells (Supplementary Fig. 7M). Collectively, these data demonstrate that pTINCR induces epithelial differentiation by regulating actin cytoskeleton dynamics through the activation of CDC42 protein and its effectors.

## Discussion

The microproteome has been largely overlooked until recently and, although some of its functions are beginning to be uncovered, much more research is needed to ascertain the extent of its biological relevance. *TINCR* was first described as a lncRNA and it has been shown to be involved in epithelial differentiation[40–42]. Here, we demonstrate that *TINCR* encodes a microprotein, pTINCR, with its own role also in epithelial differentiation. This opens the possibility that some of the functions ascribed to the lncRNA could be mediated by pTINCR. To support that our observations are due to pTINCR microprotein and not to *TINCR* lncRNA, we performed gain-of-function assays using only the pTINCR sORF and not the full-length *TINCR* lncRNA. Also, our key observations have been confirmed using a synthetic ORF (syORF) that differs by ~20% in its nucleotide sequence from the wild-type pTINCR sORF (wtORF) and changes its secondary structure. On the other hand, our pTINCR-KO cells were engineered by introducing a single nucleotide change in the start codon of pTINCR ORF. This change is predicted to have a minor effect on the folding of the *TINCR* lncRNA and did not affect its expression or transcriptional regulation. Furthermore, some of the phenotypes observed in pTINCR-KO cells were rescued with the overexpression of exogenous pTINCR. Altogether, we are confident that our observations are mediated by the microprotein and not by the lncRNA. In agreement with our results and parallel to our work, peptides derived from pTINCR have been detected by proteomics in the human stratum corneum[77], and another study has

shown that pTINCR is associated with keratinocyte proliferation[78]. However, the molecular mechanisms behind pTINCR functions were not thoroughly explored. Here, we have combined several experimental approaches, including detection by specific antibodies, to undoubtedly demonstrate the presence of pTINCR in skin but also in other epithelial tissues. In addition, we have provided evidence of its role as a tumor suppressor through the maintenance of epithelial identity. Moreover, we have performed an in-depth mechanistic analysis demonstrating that pTINCR is a UBL protein that induces epithelial differentiation, at least in part by enhancing the SUMOylation and the subsequent activation of CDC42.

We have shown that pTINCR triggers epithelial differentiation in HaCaT keratinocytes and in a set of cancer cell lines of epithelial origin, as seen by the upregulation of epithelial differentiation markers and the reorganization of actin cytoskeleton towards a cortical actin network. In addition, pTINCR reinforces cell-to-cell junctions and decreases proliferation. Furthermore, our results in teratoma formation assays evidenced the pro-differentiation role of pTINCR also in vivo. Lastly, an extensive transcriptomic analysis by RNA-seq supported these findings by showing a positive correlation between pTINCR expression and gene signatures associated with actin organization, cell-to-cell junctions, cell polarity, and epithelial differentiation. In this regard, it is worth noticing that in addition to its localization at cell-to-cell contacts and in the cytoplasm, pTINCR is also found in the nucleus, leaving open the possibility that it may have a direct role in transcriptional regulation.

By in silico analysis of its structure we observed that pTINCR belongs to the ubiquitin-like protein (UBLs) family, a family of proteins with structural similarity to ubiquitin and most of them with protein-conjugation capability[72]. Interestingly, pTINCR is a Type II UBL, which lacks the C-terminal GG residues that Type I UBLs typically used for their conjugation, as in the case of UFM, Atg12, Atg8, and UBL5. Of note, pTINCR has two overlapping SIMs and we have demonstrated that pTINCR interacts non-covalently with SUMO1 and SUMO2/3. Since the two SIM are overlapping they cannot work as two individual SIM functional domains and, consequently, pTINCR can only bind to one SUMO moiety at a given time. Importantly, we have shown that pTINCR-SUMO interaction is necessary for pTINCR stability. Furthermore, pTINCR-SIMmut does not reduce cell proliferation and does not induce remodeling of the actin cytoskeleton. Altogether, these results suggest that pTINCR function may depend on its non-covalent interaction with SUMO. In line with these results, previous evidences associate changes in cellular SUMOylation with embryonic development and cell fate stabilization[79,80], keratinocyte differentiation[81–83], and tumor suppression[84–86].

The ability to bind SUMO allows pTINCR to potentially interact with SUMOylated proteins. In this study, we uncovered an important pTINCR interactor, the small GTPase of the RHO family CDC42. Our

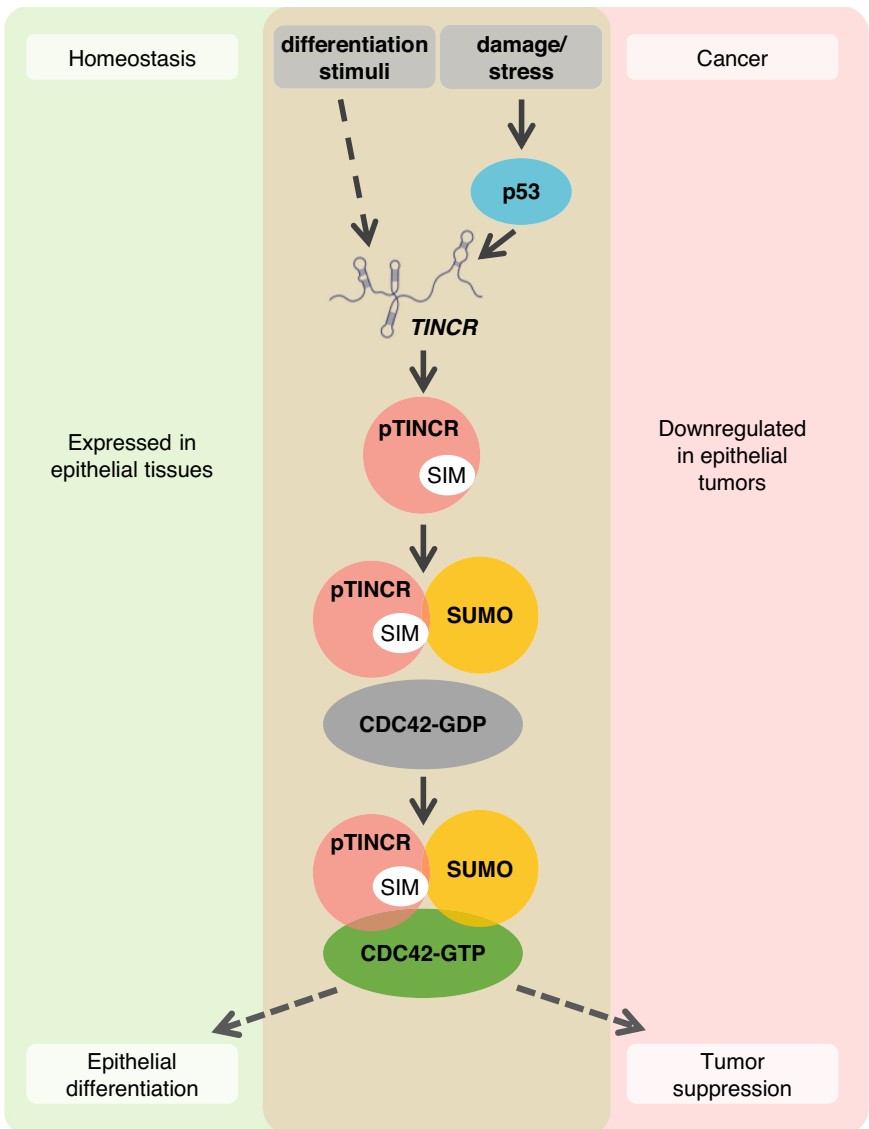

**Fig. 10 | Proposed working model for pTINCR function.** pTINCR is an UBL microprotein expressed in epithelial tissues and downregulated in cSCC. pTINCR is upregulated by pro-differentiation cues -such as calcium- and also by cellular damage in a p53 dependent manner. pTINCR interacts with SUMO through its SIM domain and modulates cell SUMOylation patterns. In particular, pTINCR increases CDC42 SUMOylation and promotes its activation, leading to epithelial differentiation and tumor suppression.

data show that pTINCR binds to CDC42, increases its SUMOylation and promotes its activation. Interestingly, mutating the pTINCR SIM-domain or treating with a SUMOylation inhibitor impair the ability of pTINCR to activate CDC42, suggesting that CDC42 SUMOylation is required for its activation. These observations are consistent with other small GTPases, such as RAC, that are also activated by SUMOylation[73]. We propose a molecular model in which pTINCR acts as a scaffold to enhance the SUMOylation of CDC42, which in turn promotes its activation. We have shown that the effect of pTINCR on SUMOylation is not a global effect but protein specific. We hypothesize that pTINCR could be promoting the SUMOylation of its interactors, such as CDC42, acting as a specificity factor that bridges UBC9 (or other complexes with SUMO-ligase activity) to their targets. This hypothesis is in line with other proteins known to regulate SUMOylation, such as the tumor suppressor p14ARF, which induces SUMOylation of a variety of p14ARF-interacting proteins[85].

CDC42 is a master regulator of cell polarity and actin dynamics[27]. In cancer, CDC42 activation has been commonly associated with oncogenic traits such as increased cell migration and invasion[87]. Our results show that pTINCR-induced CDC42 activation does not lead to lamellipodia or filopodia formation, or enhancement of mesenchymal traits. Instead, it led to cytoskeleton remodeling towards a cortical deposition of actin resembling a more differentiated state. Moreover, we have shown that treatment with CDC42 inhibitors (CASIN and ML141) impair epithelial differentiation mediated by pTINCR and by calcium. Several publications describe the importance of CDC42 for tissue differentiation, supporting its tumor suppressor role in certain contexts. During epithelial differentiation, CDC42 is involved in the establishment of an apico-basal cell polarity through the formation of tight junctions and the control of asymmetrical cell divisions[28,71]. In fact, CDC42 knock-out mice die soon after birth with severe defects in the formation of the epidermal barrier, indicating the essential role of CDC42 in epidermal differentiation[88]. Integrative genomic analyses have shown that elevated RHO and RAC signalling, but not CDC42 signalling, is a common feature in SCCs[89]. In addition, high levels of VAV2 (a guanine nucleotide exchange factor of the Rho family) in both cutaneous and head and neck SCC favors tumorigenesis through activation of RAC and RHO proteins, but not through

CDC42[90]. Altogether, we conclude that the pro-differentiation role of pTINCR is mediated, at least in part, by the SUMO-dependent activation of CDC42. In agreement, pTINCR induces a substantial increase in the phosphorylation status of several cytoskeleton-related proteins that have been widely associated with epithelial differentiation and stratification, such as Merlin[23,91,92], the p85-subunit α/γ of PI3K[93–96] or several members of the Par complex[97,98]. These last two families of proteins are downstream effectors of CDC42 and are pivotal for the establishment of the apico-basal polarity of epithelial tissues, as well as for the control of asymmetric cell division, spindle orientation and stem cell differentiation[25,76]. On the other side, we observed a significant decrease in the phosphorylation status of many proteins of the MAPK/ERK pathway, reported to be inhibited during keratinocyte differentiation[99]. The effect of many of these specific phosphorylation changes has not been reported, but our results encourage to further characterize them as potential drivers of epithelial differentiation.

*TINCR* lncRNA has been documented to be deregulated in various tumor types, and its role as an oncogene or as a tumor-suppressor seems to be tumor specific[100,101]. In esophageal SCC cells, siRNA-mediated silencing of *TINCR* represses cell proliferation and EMT features[102]. However, in other studies performed in cell lines derived from cervix, head and neck, and lung SCC, its silencing enhanced cell growth and migration[103]. In cSCC, *TINCR* lncRNA has been consistently reported to have a tumor suppressor activity[103,104]. We have corroborated the downregulation of *TINCR* lncRNA in cSCC and, moreover, we have demonstrated that pTINCR protein is also lost in human cSCC compared to healthy epidermis. Indeed, its reexpression in cSCC PDXs reduces tumor growth in vivo. Of note, a marked deposition of hyaline matrix was evident in tumors expressing pTINCR. Hyaline globules have been reported in numerous types of cancer as a rare event usually observed in benign lesions[63,105–107]. Although there is no clear consensus about its origin, this phenomenon is generally regarded as a hallmark of degeneration and apoptosis[63]. Interestingly, the presence of hyaline deposits increases in tissues after injury or in tumors after therapy[63,108,109]. Further supporting the role of pTINCR in tumor suppression, we have demonstrated that pTINCR is upregulated upon DNA damage in a p53 dependent manner. p53 is a well-known tumor suppressor and the most common mutational event in cSCC (http://p53.free.fr[62,110]), but also a known trigger of cell differentiation[59]. In fact, pTINCR-deficient cells fail to upregulate differentiation markers upon cellular damage. Finally, to expand the relevance of pTINCR in other cancer types, we performed correlation studies using public data and observed that high *TINCR* expression is associated with a favorable prognosis in patients with different tumors of epithelial origin, such as bladder, head & neck, pancreatic, stomach and lung adenocarcinomas. Taken together, these evidences strongly support that pTINCR plays a role in tumor suppression.

In summary, we have revealed that pTINCR is a previously overlooked UBL- microprotein which regulates epithelial cell identity and shows tumor suppressor activity (Fig. 10). The discovery of pTINCR as a bioactive product of an assumed lncRNA nurtures the idea of an additional level of regulation represented by lncRNA-encoded microproteins. Exploring the microproteome could provide new regulators of cell identity critical for the regulation of physiological and pathological processes, such as cancer.

## Methods

### Cell culture and treatments

HEK293T, U2OS, MCF7 and A549 cells were cultured in DMEM with GlutaMAX supplemented with 10% of fetal bovine serum (FBS) and 1% of Penicillin-Streptomycin (P/S). cSCC-patient-derived hSCC10 cell line was derived as described[48,49] and cultured in DMEM-F12 supplemented with 2% B27 (Gibco) and 1% P/S. mESC V6.4 cell line was cultured in DMEM GlutaMax supplemented with 1% Sodium Pyruvate (Invitrogen), 15% FBS, 50 mM β-mercaptoethanol, 100X non-essential amino acids (Invitrogen), 1% P/S and 1000 U/ml LIF (ESGRO, Chemicon). Low calcium (basal) or high calcium (differentiation) media were prepared by supplementing calcium-free DMEM (Gibco, #21068028) with 10% calcium-depleted FBS, 4mM L-glutamine (Invitrogen), 1% P/S, and calcium chloride (Sigma) to a final concentration of 0.03 mM (low calcium, basal media) or 2.8 mM (high calcium, differentiation media). FBS was calcium-depleted by incubation with BT Chelex 100 resin (BIO-RAD #143-2832) according to the manufacturer's instructions. HaCaT and MCF7 cells were maintained in basal media. Dedifferentiation was induced by culturing cells in basal medium for at least three weeks. Confluence was maintained lower than 70% to ensure basal conditions. Differentiation was induced by culturing with differentiation medium for the specified time (see figure legends). All cells were incubated at 37 °C, 5% $CO_2$ in a humidified incubator and routinely tested for mycoplasma contamination by PCR and confirmed negative.

Doxycycline (Sigma Aldrich) was added at 1μg/ml, when indicated, to activate Tet-inducible constructs. To induce genotoxic stress and/or p53 activation, cells were incubated with doxorubicin (1 μM), actinomycin-D (5 nM) or nutlin-3a (10 μM) for 24 h or irradiated without medium with a UV light lamp emitting at 254 nm at a rate of 50 J/m². For protein stability assays, cells were incubated with cycloheximide (100 μg/ml) for the specified time periods, 24 h after doxycycline-induced pTINCR expression. For CDC42-activation assay or F-actin immunofluorescence, hEGF (Invitrogen) was added at 200μg/ml for 5 min or 24 h, respectively. For SUMOylation inhibition, ML792 (#HY-108702, MedChemExpress) was added at 10 μM for a minimum of 16 h. For CDC42-GTP inhibition, CASIN and ML141 (#HY-12874 and #HY-12755 respectively, MedChemExpress) were added at 5 μM for 2 h minimum prior to additional treatments.

### RNA-Seq analysis of TCGA data

Transcriptomic data were extracted from TCGA (database https://portal.gdc.cancer.gov/) in the form of count tables or from Das Mahapatra et al. study (cSCC cases)[44] in the form of fastq files. In this last case, paired-end reads from RNA-Seq were aligned using Tophat v.2.1.0 to the human genome (hg19). Predicted transcripts from Ensembl database (release 87) were analyzed. Differentially expressed genes (DEG) were identified using HTSeq + DESeq2 v. 1.24.0[55,111].

### Ribosome profiling analysis

We adapted a computational approach to identify translated sORFs[112] by analyzing a public ribosome profiling dataset from mouse skin (GSE83332[19]). In brief, read adapters were trimmed and reads mapping to annotated ribosomal and transfer RNAs were filtered out. The resulting reads were mapped to the assembled mouse genome (mm10) using Bowtie2 v2.3.4.3 with default parameters. Next, mapped reads from experimental replicates were merged and we used the ribORF algorithm[113] to predict translated sORFs in *TINCR* lncRNA with significant read uniformity and frame periodicity (score ≥ 0.7), as this feature is indicative of ORF active translation.

### In vitro transcription/translation

lncRNA *TINCR* CDS was cloned in a pcDNA-3.1 vector under the control of the T7 promoter and incubated for 1 h with rabbit reticulocyte-coupled transcription/translation system (Promega) in the presence of [35 S] methionine. After incubation, translated product was resolved by 15% SDS-polyacrylamide gel electrophoresis (PAGE) and detected by autoradiography.

### LC-MS/MS identification of pTINCR microprotein

Identification of pTINCR peptides was performed using publicly available data of organotypic skin cell cultures from Elias et al[45]. In

brief, dataset with identifier PDX014088 was downloaded from PRIDE repository (https://www.ebi.ac.uk/pride/archive). Raw files were processed with Proteome Discoverer 2.2 (Thermo Fisher Scientific) using the standard settings against a human protein database (UniProtKB/Swiss-Prot, July 2018, 20,373 sequences), supplemented with contaminants and the pTINCR sequences. Carbamidomethylation of cysteines was set as a fixed modification whereas methionine oxidation and N-term acetylation were variable protein modifications. The minimal peptide length was set to 6 amino acids and a maximum of two tryptic missed cleavages were allowed. The results were filtered at 0.01 FDR (peptide and protein level).

## Histology and immunohistochemistry

Immunostaining was performed on paraffin-embedded mouse and human tissues. In brief, paraffin blocks were sliced into 5-μM thick sections, deparaffinized with xylene (Fisher Scientific, Waltham, MA, USA) and rehydrated with decreasing concentrations of ethanol in water. For immunohistochemistry, paraffin sections underwent antigenic exposure process into the Discovery Ultra (Ventana) system with CC1 buffer for 64 min at 95 °C. Anti-pTINCR antibody was incubated for 1 h at room temperature (Supplementary Table 7). Next, slides were incubated with the secondary antibody Discovery UltraMap anti-Rabbit HRP (Ventana). H&E staining was performed on 5 μm paraffin sections in a Robust carousel tissue stainer (Slee Medical) according to common method.

## Cloning procedures

pTINCR ORF was synthesized (IDT technologies) fused with a flexible linker (GGGGSGGGGSGGGGS) and an HA-Tag epitope at the C-terminal or N-terminal part of the microprotein and flanked by EcoRI enzyme restrictions sites at both ends. After enzymatic digestion, constructs were ligated into the pENTR1A vector. pTINCR-SIM mut plasmid was purchased directly cloned into the pDONR201 plasmid, in-frame with an HA-tag epitope in its C-terminal (Proteogenix). Subsequently, pTINCR-HA, HA-pTINCR and pTINCR-SIMmut constructs (Supplementary Table 2) were obtained by recombining donor vectors with the lentiviral inducible system pINDUCER20 (Invitrogen) using the Gateway Cloning Technology, following manufacturer's instructions. As specified, some experiments were reproduced using a pTINCR synthetic ORF (syORF), generated by mutating the pTINCR ORF in pTINCR-HA and pTINCR SIMmut constructs (Supplementary Table 2).

## Lentiviral transduction

HEK293T cells were transfected with 5 μg of specific plasmid and 5 μg of packaging vectors (pLP-1, pLP-2, pLP-VSVG, Invitrogen) using Fugene HD (Promega) following the manufacturer's instructions. Viral supernatants were collected twice a day on two consecutive days, filtered through a 0.45 μm syringe filter, supplemented with of 8 μg/ml of polybrene and used to infect HaCaT, A549, MCF7, hSCC10 or mESCs. Successfully infected cells were established by geneticin selection.

## Analysis of mRNA levels

Total RNA was extracted with Trizol (Invitrogen) following the manufacturer's protocol. Genomic DNA was cleaned up and retro-transcription performed using the iScript gDNA Clear cDNA Synthesis Kit (BioRad). Gene expression was analyzed by RT-qPCR using PowerUp SYBR Green Master Mix (Thermo Fisher Scientific) in the 7900HT Fast Real-Time PCR System (Applied Biosystems). Gene-specific primers are listed in Supplementary Table 5, 6. Cycle threshold (Ct) values were normalized to GAPDH.

## Western blot

Cells were scrapped and homogenized in medium-salt lysis buffer (150 mM NaCl, 50 mM Tris pH 8, 1% NP40 and protease inhibitors cocktail) and concentrations determined using the Pierce™ BCA Protein Assay Kit (Thermo Fisher). Lysates were loaded in acrylamide gels for electrophoresis in Tris-Glycine SDS Running Buffer. Primary antibodies were incubated overnight at 4 °C. Secondary HRP-conjugated antibodies were incubated the following day for 1 h at room temperature, and ECL Prime Western Blotting Detection Reagent (Fisher Scientific) was used as a chemiluminescent reagent for protein detection. Antibodies and dilutions are listed in Supplementary Table 7.

## Subcellular fractionation

Cells were homogenized in Buffer A (HEPES pH 7.8 (10 mM), MgCl2 (1.5 mM), KCl (10 mM) and DTT). The homogenate was incubated on ice for 10 min and then 10% Triton-X was added to favor cellular disruption. Samples were centrifuged at 11,300 g for 1 min at 4 °C and supernatant (Cytoplasmic extract, CE) was separated from the pellet (Nuclear extract, NE). CE supernatant was washed with Buffer B (HEPES pH 7.8 (0.3 mM), MgCl2 (1.4 mM) and KCl (30 mM)), followed by centrifugation at 15,000 g for 15 min at 4 °C. New supernatant was used as final cytoplasmic extract and pellet was disrupted in SDS lysis buffer (Tris-HCl pH 8.0 (50 mM), EDTA (1 mM), SDS (2%)) to obtained final cellular membranes extract. NE pellet was washed with Buffer B and disrupted using Buffer C (HEPES pH 7.8 (20 mM), MgCl2 (1.5 mM), NaCl (0.42 mM), EDTA (0.2 mM), glycerol (25%) and DTT). After 30 min of iced-incubation and centrifugation at 15,000 g for 15 min at 4 °C, the supernatant was used as the final nuclear extract.

## Immunofluorescence

Cells were seeded in fibronectin-coated coverslips (Sigma Aldrich). When desired, cells were fixed in 2% paraformaldehyde for 15 min and permeabilized with 0.5% Triton X-100 for 10 min at room temperature. Blocking step was made in 3% Bovine Serum Albumin (BSA) for 1 h. Cells were incubated overnight at 4 °C with the primary antibody diluted in blocking buffer. Next day, secondary antibodies were incubated for 1 h at room temperature in the dark. Antibodies and dilutions are listed in Supplementary Table 7. Finally, cells were mounted in Prolong Mounting Medium with DAPI (Invitrogen) and images were taken in a Nikon Eclipse Ti-E inverted microscope system. For quantification, at least 200 cells per staining were evaluated using ImageJ software.

## Cell proliferation analysis

Cells were seeded into 24-well cell culture plates at density $5 \times 10^3$ cells/well. Every two days, cells were trypsinised and counted using a Neubauer chamber.

## Generation of pTINCR-KO cells

HaCaT and MCF7 cells were transfected with pSpCas9(BB)−2A-Puro plasmid, which contains a sgRNAs (AGCCGGGCGGGCGCCATGGAGGG) design to target the first exon of TINCR gene loci. Twenty-four hours after transfection, puromycin was added to select for infected cells. Isolation of pTINCR-KO single colonies was assessed by serial limiting dilution in 96-multiwell plate. Successfully CRISPR-CAS9 edited clones were screened by genotyping PCR.

## RNA sequencing

Cells were cultured and treated with doxycycline for microprotein expression induction during the specified experimental time periods. Total RNA was isolated with Trizol following manufacturer's protocol. RNA quantity and purity were measured with the Nanodrop spectrophotometer (Thermo Scientific) and 1 μg of total RNA was processed for sequencing analysis. RNA integrity, determined by the RNA integrity number (RIN), was determined with the 2100 Bioanalyzer (Agilent Technologies Inc., Santa Clara, CA). Cytoplasmatic and mitochondrial ribosomal RNAs were depleted using the RiboZero Magnetic Gold Kit

(Illumina Inc). rRNA-depleted samples were fragmented, cDNA was synthesized and converted into sequentiable libraries using the TruSeq Stranded Total RNA kit protocol (Illumina Inc.). The size and quality of the libraries were assessed with a High Sensitivity DNA Bioanalyzer assay (Agilent Technologies Inc., Santa Clara, CA). Libraries were sequenced in a The NextSeq 500™ (Illumina Inc.), with a read length of 2x76bp. On average, 77 million paired-end reads were generated per sample. Image analysis, base calling and quality scoring of the run were processed using the manufacturer's software Real Time Analysis (RTA 1.18.64) and followed by generation of FASTQ sequence files by CASAVA.

### RNAseq data analysis

Paired end reads were aligned to the hg38 human genome with STAR (v2.5.2b) and default parameters. Sambamba (v0.6.7) was used to convert to bam and sort resulting sam files. All subsequent analyses were performed in the R programming environment (https://www.R-project.org) unless otherwise stated. Count matrices were generated with the Rsubread package[114] using the inbuilt annotation hg38. Genes were annotated using biomaRt (vGRCh38.p12). Normalization and contrasts were performed using the DESeq2[55] R package. Time points were compared against 0 h within the control and pTINCR-overexpressing condition.

In order to find genes whose expression was significantly associated with time we applied the runImpulseDE2 function from the ImpulseDE2[54] R package on the count matrix generated with Rsubread. The biological replicate was included as a technical confounder. Genes with adjusted p-values lower than 0.001 were selected for downstream analyses. The matrix containing the fold changes resulting from the comparisons against 0 h was filtered for those genes with significant association with time according to ImpulseDE. The cmeans function from the e107 R package was used to find clusters of genes with similar expression patterns along time. The centers parameter was fixed at 6 after manual inspection. The ggplot2 (https://ggplot2.tidyverse.org), viridis, ggExtra and gridExtra R packages were used for plotting resulting clusters. Genes were classified in a given cluster if their membership probabilities were larger than 0.5. Functional enrichment on the resulting gene clusters was performed using a hypergeometric test against the Gene Ontology database[115], KEGG pathways[116] and Broad Hallmarks gene sets[117].

Networks of overlapping gene ontology terms were computed using the ClueGO (v2.5.7) module of Cytoscape (v3.9.1) with default parameters.

Furthermore, the interaction between treatment and time was computed for times 0 h and 21 days. Functional enrichment of the interaction coefficients was performed using a rotation-based methodology. The ROAST algorithm (v0.99.1) as implemented in the R package limma was used to represent the null distribution. The max-mean enrichment statistic proposed in[118], under restandarization, was considered for competitive testing.

### Human cSCC samples collection and processing

A historical cohort of fifty-one patients diagnosed with cSCC from January 2009 to August 2010 was included in the present study provided by the Tumor Bank of the Vall d'Hebron University Hospital Biobank. For histological examination, H&E staining of formalin-fixed and paraffin-embedded (FFPE) tumor sample was performed. All cases were evaluated independently by two pathologists (SRYC, OM).

For immunohistochemistry studies of pTINCR and p53, whole slide FFPE tissue sections of 5 μm of selected samples were stained as previously described. All cases were evaluated independently by an one expert dermatopathologist and one trained Molecular Biologist blinded for patient groups, considering the percentage of positive cells and intensity of the staining, which was assessed semi-quantitatively. Final results were obtained utilizing the average of

the two values. Whenever a major discrepancy was observed between both observers, the cases were discussed using a multi-headed microscope.

For mutational profiling, FFPE tissue sections of 10 μm were stained with H&E and examined by a pathologist to select for a minimum tumor content of 20% as a requirement for further processing.

### Amplicon-seq analysis

DNA was extracted from $5 \times 10$ μm sliced FFPE sections using the Maxwell FFPE Tissue LEV DNA Purification Kit (Promega), according to the manufacturer's instructions. DNA quality and concentration were determined by fluorometric quantification using Qubit Fluorimeter and Qubit dsDNA BR Assay Kit (Life Technologies, Carlsbad, CA).

Tumor DNA was sequenced with an in-house developed amplicon-sequencing panel of over 60 genes and 1330 primer pairs targeting frequent mutations in oncogenes and several tumor suppressors. A total of 500 ng of DNA from each tissue sample were used for library preparation, according to our established protocols. An initial multiplex-PCR with a proof-reading polymerase was performed on all samples. Indexed libraries were pooled and sequenced in a MiSeq instrument ($2 \times 100$) at an average coverage of 1000X. Initial alignment was performed with BWA (v0.7.17) after primer sequence clipping and variant calling was done with the GATK Unified Genotyper (v3.4.0) and VarScan2 (v2.4.3) followed by ANNOVAR annotation (v.annovar_180416). Mutations were called at a minimum 3% allele frequency. SNPs were filtered out with dbSNP and 1000 genome datasets. All detected variants were manually checked. Technical quality parameters are provided in Supplementary Table 8.

### ChipSeq analysis

Data was extracted from GSE58506 and GSE100292, aligned with GRCh37/hg19 and visualized with the Integrative Genomics Viewer (IGV) (v2.13.2).

### Non-covalent SUMO binding assay

Glutathione S-transferase (GST) pull-down experiments were performed by incubating protein extracts from U2OS cells stably expressing pTINCR-HA or pTINCR SIMmut treated with Doxycycline for 48 h, together with GST, GST-SUMO1 or GST-SUMO2/3 bound onto glutathione-sepharose beads for 2 h at 4 °C. Then, beads were washed and bound proteins were eluted, separated on 14% SDS-PAGE and detected by Western-blot.

### Interactome analysis and validation

Cells were lysed in a buffer containing 50 mM Tris-Hcl pH 7.5-8, 150 mM NaCl, 1%Triton X-100 and protease inhibitors, and homogenized for 30 min in a rotor wheel. Lysates (3 mg) were immuno-precipitated with 5μg of monoclonal HA-Tag antibody overnight. Immunocomplexes were collected using PureProteome™ Protein A Magnetic Beads (MERCK) and eluted by competition with a synthetic HA peptide (Sigma). Eluate was digested with trypsin and analyzed by liquid chromatography-mass spectrometry on an LTQ Orbitrap Velos instrument (ThermoFisher). Progenesis ® QI for proteomics software v3.0 (Nonlinear dynamics, UK) was used for MS data analysis using default settings. The LC-MS runs were automatically aligned to an automatically selected reference sample with manual supervision Peak lists were generated from Progenesis and loaded to Proteome Discoverer v2.1 (Thermo Fisher Scientific) for protein identification. Proteins were identified using Mascot v2.5 (Matrix Science, London UK) to search the SwissProt database (2018_11, taxonomy restricted to human proteins, 20,240 sequences). Significance threshold for the identifications was set to $p < 0.05$, minimum ions score of 20. Statistical analysis was performed using Progenesis software. Proteins displaying greater than 2-fold change, and $p < 0.05$ (T-test) between IP and

control groups were considered significantly differential. Interactions were analyzed using CRAPome algorithm[119]. For validation of the results, U2OS cells were co-transfected with pTINCR-HA and CDC42-Flag plasmids and immunoprecipitation was performed as explained before. SDS-PAGE was used to visualized anti-HA pTINCR and anti-Flag CDC42 proteins.

### CDC42 activation assay

The activation of CDC42 (CDC42-GTP) was estimated using the Cdc42 activation kit (Cytoskeleton Inc.). Briefly, after 4 days of pTINCR doxycycline induction cells were washed with cold PBS, quickly lysed using provided lysis buffer and concentrations determined using the Pierce™ BCA Protein Assay Kit (Thermo Fisher). GTP-bound CDC42 was pulled-down by GST-p21 binding domain (PBD) and detected with anti-CDC42 antibody (Supplementary Table 7) according to the manufacturer's instructions. Positive control was the result of treating control cells with EGF for 5 min prior to the procedure.

### SUMO conjugation assays

SUMO conjugation assays were performed by transfecting U2OS stably expressing pTINCR cell line with HIS6-SUMO2 protein and, when required, co-transfected CDC42 Flag-tagged plasmids. After 48 h of doxycycline treatment, cells were lysed with 6 M guanidinium-HCl, 0.1 M $Na_2HPO_4/NaH_2PO_4$ buffer. Then, lysates were mixed with $Ni^{2+}$-nitrilotriacetic acid-agarose beads and incubated for 2 h at room temperature. After washing, beads were resuspended in Laemmli buffer and purified proteins were subjected to SDS-PAGE electrophoresis as indicated above.

### Cell migration assay

A549 cells were seeded and culture until confluence was reached. Then, a pipette tip was used to scratch the surface of cell monolayer, forming a wound. An Olympus CellR microscope equipped with a Hamamatsu C9100 camera was used to follow the closure of the wound up to 48 h.

hSCC10 cells were seeded in a Corning cell culture insert (Boyden chamber). Cells were allowed to migrate through for 24 h at 37 °C. After that time, culture inserts were fixed in methanol for 5 min and stained using Crystal Violet during 20 min at room temperature. Pictures were taken using an Olympus CellR microscope. Cells were counted using ImageJ.

### Cell invasion assay

$2 \times 10^4$ cells were seeded in a Corning cell culture insert (Boyden chamber) coated with a layer of Matrigel (Corning). Cells were allowed to invade through the membrane for 24 h at 37 °C. After that time, culture inserts were fixed in methanol for 5 min and stained using Crystal Violet during 20 min at room temperature. Pictures were taken using an Olympus CellR microscope. Cells were counted using ImageJ.

### Human cytoskeleton phospho-array

Phosphorylation-specific antibody microarray (Fullmoon Biosystems Inc.) was used to determine the up- and down-regulated proteins in hSCC10 cells after 1 day of pTINCR induction. The array contains 141 site-specific antibodies against phosphorylated- and unphosphorylated-proteins involved in cytoskeletal pathways, each replicated six times. Actin and GAPDH were used as controls. The assay was performed following manufacturer's protocol.

### Animal procedures

Mice were kept in enriched shelters in a pathogen-free facility with 12 h/12 h light/dark cycle (from 08:00am to 08:00 pm) with ad libitum access to food and water. Housing conditions were maintained at an average temperature of 22 ± 2 °C and humidity 55 ± 10%. Mice were checked daily for general condition and human endpoints.

For subcutaneous teratomas and patient-derived xenografts (PDXs), mESCs V6.4 or hSCC10 cells lines, respectively, expressing either pTINCR HA-tagged constructs or control vector were trypsinized and $1 \times 10^6$ cells were subcutaneously injected into the flanks of 8-week-old immunocompromised male NMRI mice (RjHan:NMRI). Tumor growth was monitored twice a week using the formula height × width × width × (3,1416/6) and animals were sacrificed when tumors reached 1.7 cm³ or before at the indicated time. Doxycycline was administered in the drinking water at 1 mg/ml supplemented with 7.5% of sucrose every two days throughout all the experiments.

### Generation of Kaplan−Meier plots

Kaplan-Meier (KM) plots were generated for survival analysis using KM plotter database (http://kmplot.com/analysis), a website database based on resources from TCGA database. The final prognostic KM plots were presented with a hazard ratio (HR), 95% confidence interval (CI) and log-rank $P$ value. $P$ value < 0.05 was considered statistically significant

### Statistics and reproducibility

Data are expressed as the mean ± standard deviation (SD) or the mean ± standard error of the mean (SEM), as specified. Differences between groups were analyzed using unpaired T-TEST, Multiple T-TEST correct for multiple comparison, one or two-way ANOVA with multiple correction, Fisher exact test or non-parametric Kruskal-Wallis test, as specified. Corrections for multiple comparison were performed when necessary: one and two-way ANOVA with Dunnett method, multiple comparison T-TEST with Holm-Sidak method and Kruskal−Wallis test with Dunn's test. All statistical tests were two-sided and performed using GraphPad Prism (v8.4.0, GraphPad Software Inc., San Diego, CA, USA). The experiments shown in this manuscript were repeated independently at least twice obtaining similar results, unless specified.

### Ethical statement

The animal studies in this work comply with the European, Spanish, and Catalan Regulations for the Protection of Vertebrate Animals used for Experimental and other Scientific Purposes (Directive 2010/63; Spanish BOE RD 53/2013; Catalan DOGC 214/1997). All studies were carried out in the "Lab Animal Service Campus Vall d'Hebron (LAS-CVH)", registered and accredited at the Departament de Medi Ambient i Habitatge by Generalitat of Catalonia government with register number B9900062. The experiments performed for this manuscript were linked to a project approved by Vall d'Hebron Ethics Committee, and the Commission of Animal Experimentation of Generalitat of Catalonia government.

Studies involving human samples have been performed in accordance with the Declaration of Helsinki. All patient samples studied have the approval of the Ethics Committee for Research of the Vall d'Hebron University Hospital (PR research project PR research project (AG) 191/2019)). Samples recruitment was not done specifically for our study.

### Materials availability

All the unique reagents described in this work will be made available according to the guidelines of the journal.

### Reporting summary

Further information on research design is available in the Nature Portfolio Reporting Summary linked to this article.

## Data availability

The RNA-seq data generated for this publication has been deposited in NCBI's Gene Expression Omnibus (https://www.ncbi.nlm.nih.gov/geo)

and are accessible through GEO Series accession number GSE175463. The mass spectrometry proteomics data have been deposited to the ProteomeXchange Consortium via the PRIDE partner repository and are accessible with the dataset identifier PXD026181. Published datasets included in this study: PDX014088 (https://www.ebi.ac.uk/pride/archive/), GSE139505, GSE83332, GSE58506, GSE100292. Transcriptomic data was downloaded from GTex portal (https://gtexportal.org/home/). Figures with associated raw data are Fig. 5, Fig. 9b and Supplementary Fig. 4 (GSE175463), Fig. 9a and Supplementary Table 3 (PXD026181), Fig. 1a (GSE139505), Fig. 1c (GSE83332), Fig. 1g and Supplementary Fig. 1a (PDX014088) and Supplementary Fig. 5E and F (GSE58506, GSE100292), Fig. 1b (GTex). All other data that support the findings of this study are available within the article, its supplementary information, or Source Data file. Source data are provided with this paper.

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

## Acknowledgements

The authors thank VHIO Proteomics, Molecular Oncology and Genomics Core Facilities for technical assistance. We are grateful to Manuel Serrano for providing several reagents, advice and critical discussion on the manuscript. We also thank Alonso García and Raquel Pérez for their help in processing and analyzing digital images, Gemma Serra and Sandra Peiró for their assistance with subcellular fractionation and immunoprecipitation experiments, Sara Arce and Joaquín Mateo for providing several reagents during the development of critical experiments of this manuscript, and Juan Angel Recio for his help with the cSCC cohort. We are immensely grateful to all the members of the Abad lab for generating the know-how for the identification of novel sORFs, for the critical reading on the manuscript and in general for their constant support to this project. Work in the Abad lab is supported by VHIO, Fero Foundation, La Caixa Foundation, Asociación Española Contra el Cancer (AECC), La Mutua Foundation and by grants from the Spanish Ministry of Science and Innovation (SAF2015-69413-R; RTI2018-102046-B-I00). M.A. was recipient of a Ramón y Cajal contract from the Spanish Ministry of Science and Innovation (RYC-2013-14747). O.B. is recipient of a FPI-AGAUR fellowship from Generalitat de Catalunya. We also acknowledge funding from grant PGC2018-094091-B-I00 from the Spanish Government.

## Author contributions

O.B. designed and performed most of the experiments and data analysis, contributed to discussion and co-wrote the manuscript. M.M. performed a substantial number of experiments and contributed to discussion and writing. S.V.F. and C.R. performed most of the experiments related with SUMO. M.G. and L.P generated the pTINCR-KO cell lines and provided general technical assistance throughout the project. L.L.S. and P.M. generated and provided the patient-derived cSCC cell lines. L.Q. and I.V. analyzed cSCC transcriptomic data and correlated TINCR expression with clinical data in multiple tumor types. O.M., N.C., P.N., and S.R.Y.C. performed the histopathological analyses. J.R.O. and

M.M.A. analyzed the Ribo-seq. P.X.E. and J.M. analyzed the MS data on organotypic skin cultures. T.V.T. analyzed the p53 ChIP-seq experiments. A.V. performed generated the libraries and performed the RNA-seq on cSCC cells. C.S.O.A. analyzed the RNA-seq on cSCC cells. M.A. designed and supervised the study, secured funding, analyzed the data, and wrote the manuscript. All authors discussed the results and commented on the manuscript.

## Competing interests
The authors declare no competing interests.

## Additional information

[1]Cellular Plasticity and Cancer Group, Vall d'Hebron Institute of Oncology (VHIO), Barcelona, Spain. [2]Centro de Investigación en Medicina Molecular (CIMUS), Universidad de Santiago de Compostela and Instituto de Investigaciones Sanitarias (IDIS), Santiago de Compostela, Spain. [3]Oncobell Program, Bellvitge Biomedical Research Institute (IDIBELL), Barcelona, Spain. [4]Instituto de Biomedicina y Biotecnología de Cantabria, Universidad de Cantabria - CSIC, Santander, Spain. [5]Department of Pathology, Vall d'Hebron University Hospital, Translational Molecular Pathology, Vall d'Hebron Institute of Research (VHIR), Universitat Autònoma de Barcelona and Spanish Biomedical Research Network Centre in Oncology (CIBERONC), Barcelona, Spain. [6]Cardiovascular and Metabolic Sciences, Max Delbrück Center for Molecular Medicine in the Helmholtz Association (MDC), Berlin, Germany. [7]Proteomics Unit, Spanish National Cancer Research Centre (CNIO), Madrid, Spain. [8]Molecular Oncology Group, Vall d'Hebron Institute of Oncology (VHIO), Barcelona, Spain. [9]Bioinformatics-Biostatistics Unit, Institute for Research in Biomedicine (IRB Barcelona), Barcelona Institute of Science and Technology (BIST), Barcelona, Spain. [10]IMIM - Hospital del Mar Medical Research Institute, Barcelona, Spain. [11]Catalan Institution for Research and Advanced Studies (ICREA), Barcelona, Spain. [12]Non-colorectal Gastrointestinal Cancer Translational Research Group, Vall d'Hebron Institute of Oncology (VHIO), Barcelona, Spain. [13]Cancer Genomics Group, Vall d'Hebron Institute of Oncology (VHIO), Barcelona, Spain. [14]Departamento de Biología Molecular y Celular, Centro Nacional de Biotecnología (CNB), CSIC, Darwin 3, 28049 Madrid, Spain. ✉e-mail: mabad@vhio.net

