## [Peer Review File · Nature Communications]

pTINCR microprotein promotes epithelial differentiation and suppresses tumor growth through CDC42 SUMOylation and activationEditorial Note: This manuscript has been previously reviewed at another journal that is not operating a transparent peer review scheme. This document only contains reviewer comments and rebuttal letters for versions considered at *Nature Communications*.

REVIEWERS' COMMENTS

Reviewer #1 (Remarks to the Author):

The authors have addressed my concerns and the paper is suitable for publication in Nature Communications.

Reviewer #2 (Remarks to the Author):

The major limitations of the study have been addressed or clearly discussed in the revised text, and the study is appropriate for publication.

Reviewer #3 (Remarks to the Author):

The authors have addressed the remaining concerns I had regarding the pTINCR-SUMO interaction through the SIM domains.

Overall, a rather exciting manuscript on the characterisation of TINCRlncRNA as tumour suppressor through expression of pTINCR and its function as UBL.

Nature Communications manuscript NCOMMS-22-30232-T
Point-by-point response to the reviewers

[AUTHORS]

We would like to thank the reviewers for their time in evaluating our work once again and for their positive comments.

Reviewers' comments:

Reviewer #1 (Remarks to the Author):

The authors have addressed my concerns and the paper is suitable for publication in Nature Communications.

[AUTHORS] Many thanks. We are glad that the reviewer considers the paper suitable for publication.

Reviewer #2 (Remarks to the Author):

The major limitations of the study have been addressed or clearly discussed in the revised text, and the study is appropriate for publication.

[AUTHORS] We are glad to know that the reviewer is satisfied with the revised version of the manuscript and find it appropriate for publication.

Reviewer #3 (Remarks to the Author):

The authors have addressed the remaining concerns I had regarding the pTINCR-SUMO interaction through the SIM domains.

Overall, a rather exciting manuscript on the characterisation of TINCRlncRNA as tumour suppressor through expression of pTINCR and its function as UBL.

[AUTHORS] We thank the reviewer for his positive comment. We are happy to know that his last concern has been addressed.